# HARNESSING STRUCTURES FOR VALUE-BASED PLANNING AND REINFORCEMENT LEARNING

**Yuzhe Yang**[*], **Guo Zhang**[*], **Zhi Xu**[*], **Dina Katabi**
Computer Science and Artificial Intelligence Lab
Massachusetts Institute of Technology
`{yuzhe, guozhang, zhixu, dk}@mit.edu`

## ABSTRACT

Value-based methods constitute a fundamental methodology in planning and deep reinforcement learning (RL). In this paper, we propose to exploit the underlying structures of the state-action value function, i.e., $Q$ function, for both planning and deep RL. In particular, if the underlying system dynamics lead to some global structures of the $Q$ function, one should be capable of inferring the function better by leveraging such structures. Specifically, we investigate the *low-rank* structure, which widely exists for big data matrices. We verify empirically the existence of low-rank $Q$ functions in the context of control and deep RL tasks. As our key contribution, by leveraging Matrix Estimation (ME) techniques, we propose a general framework to exploit the underlying low-rank structure in $Q$ functions. This leads to a more efficient planning procedure for classical control, and additionally, a simple scheme that can be applied to value-based RL techniques to consistently achieve better performance on "low-rank" tasks. Extensive experiments on control tasks and Atari games confirm the efficacy of our approach.

## 1 INTRODUCTION

Value-based methods are widely used in control, planning, and reinforcement learning (Gorodetsky et al., 2018; Alora et al., 2016; Mnih et al., 2015). To solve a Markov Decision Process (MDP), one common method is value iteration, which finds the optimal value function. This process can be done by iteratively computing and updating the state-action value function, represented by $Q(s, a)$ (i.e., the $Q$-value function). In simple cases with small state and action spaces, value iteration can be ideal for efficient and accurate planning. However, for modern MDPs, the data that encodes the value function usually lies in thousands or millions of dimensions (Gorodetsky et al., 2018; 2019), including images in deep reinforcement learning (Mnih et al., 2015; Tassa et al., 2018). These practical constraints significantly hamper the efficiency and applicability of the vanilla value iteration.

Yet, the $Q$-value function is intrinsically induced by the underlying system dynamics. These dynamics are likely to possess some structured forms in various settings, such as being governed by partial differential equations. In addition, states and actions may also contain latent features (e.g., similar states could have similar optimal actions). Thus, it is reasonable to expect the structured dynamic to impose a *structure* on the $Q$-value. Since the $Q$ function can be treated as a giant matrix, with rows as states and columns as actions, a structured $Q$ function naturally translates to a *structured Q* matrix.

In this work, we explore the *low-rank* structures. To check whether low-rank $Q$ matrices are common, we examine the benchmark Atari games, as well as 4 classical stochastic control tasks. As we demonstrate in Sections 3 and 4, more than 40 out of 57 Atari games and all 4 control tasks exhibit low-rank $Q$ matrices. This leads us to a natural question: How do we leverage the low-rank structure in $Q$ matrices to allow value-based techniques to achieve better performance on "low-rank" tasks?

We propose a generic framework that allows for exploiting the low-rank structure in both classical planning and modern deep RL. Our scheme leverages Matrix Estimation (ME), a theoretically guaranteed framework for recovering low-rank matrices from noisy or incomplete measurements (Chen & Chi, 2018). In particular, for classical control tasks, we propose Structured Value-based Planning (SVP). For the $Q$ matrix of dimension $|\mathcal{S}| \times |\mathcal{A}|$, at each value iteration, SVP randomly updates a small portion of the $Q(s, a)$ and employs ME to reconstruct the remaining elements. We show that

---

[*] Equal contribution. Code is available at: `https://github.com/YyzHarry/SV-RL`

(a) Vanilla value iteration  (b) Vanilla value iteration  (c) Online reconstruction  (d) Online reconstruction

Figure 1: The approximate rank and MSE of $Q^{(t)}$ during value iteration. (a) & (b) use vanilla value iteration; (c) & (d) use online reconstruction with only 50% observed data each iteration.

planning problems can greatly benefit from such a scheme, where fewer samples (only sample around 20% of $(s, a)$ pairs at each iteration) can achieve almost the same performance as the optimal policy.

For more advanced deep RL tasks, we extend our intuition and propose Structured Value-based Deep RL (SV-RL), applicable for deep $Q$-value based methods such as DQN (Mnih et al., 2015). Here, instead of the full $Q$ matrix, SV-RL naturally focuses on the "sub-matrix", corresponding to the sampled batch of states at the current iteration. For each sampled $Q$ matrix, we again apply ME to represent the deep $Q$ learning target in a structured way, which poses a low rank regularization on this "sub-matrix" throughout the training process, and hence eventually the $Q$-network's predictions. Intuitively, as learning a deep RL policy is often noisy with high variance, if the task possesses a low-rank property, this scheme will give a clear guidance on the learning space during training, after which a better policy can be anticipated. We confirm that SV-RL indeed can improve the performance of various value-based methods on "low-rank" Atari games: SV-RL consistently achieves higher scores on those games. Interestingly, for complex, "high-rank" games, SV-RL performs comparably. ME naturally seeks solutions that balance low rank and a small reconstruction error (cf. Section 3.1). Such a balance on reconstruction error helps to maintain or only slightly degrade the performance for "high-rank" situation. We summarize our contributions as follows:

- We are the first to propose a framework that leverages matrix estimation as a general scheme to exploit the low-rank structures, from planning to deep reinforcement learning.

- We demonstrate the effectiveness of our approach on classical stochastic control tasks, where the low-rank structure allows for efficient planning with less computation.

- We extend our scheme to deep RL, which is naturally applicable for value-based techniques. Across a variety of methods, such as DQN, double DQN, and dueling DQN, experimental results on all Atari games show that SV-RL can consistently improve the performance of value-based methods, achieving higher scores for tasks when low-rank structures are confirmed to exist.

## 2 WARM-UP: DESIGN MOTIVATION FROM A TOY EXAMPLE

To motivate our method, let us first investigate a toy example which helps to understand the structure within the $Q$-value function. We consider a simple deterministic MDP, with 1000 states, 100 actions and a deterministic state transition for each action. The reward $r(s, a)$ is randomly generated first for each $(s, a)$ pair, and then fixed throughout. A discount factor $\gamma = 0.95$ is used. The deterministic nature imposes a strong relationship among connected states. In this case, our goal is to explore: (1) what kind of structures the $Q$ function may contain; and (2) how to effectively exploit such structures.

**The Low-rank Structure** Under this setting, $Q$-value could be viewed as a $1000 \times 100$ matrix. To probe the structure of the $Q$-value function, we perform the standard $Q$-value iteration as follows:

$$Q^{(t+1)}(s, a) = \sum_{s' \in \mathcal{S}} P(s'|s, a)\big[r(s, a) + \gamma \max_{a' \in \mathcal{A}} Q^{(t)}(s', a')\big], \quad \forall (s, a) \in \mathcal{S} \times \mathcal{A}, \quad (1)$$

where $s'$ denotes the next state after taking action $a$ at state $s$. We randomly initialize $Q^{(0)}$. In Fig. 1, we show the approximate rank of $Q^{(t)}$ and the mean-square error (MSE) between $Q^{(t)}$ and the optimal $Q^*$, during each value iteration. Here, the approximate rank is defined as the first $k$ singular values (denoted by $\sigma$) that capture more than 99% variance of all singular values, i.e., $\sum_{i=1}^{k} \sigma_i^2 / \sum_j \sigma_j^2 \geq 0.99$. As illustrated in Fig. 1(a) and 1(b), the standard theory guarantees the

convergence to $Q^*$; more interestingly, the converged $Q^*$ is of low rank, and the approximate rank of $Q^{(t)}$ drops fast. These observations give a strong evidence for the intrinsic low dimensionality of this toy MDP. Naturally, an algorithm that leverages such structures would be much desired.

**Efficient Planning via Online Reconstruction with Matrix Estimation** The previous results motivate us to exploit the structure in value function for efficient planning. The idea is simple:

*If the eventual matrix is low-rank, why not enforcing such a structure throughout the iterations?*

In other words, with the existence of a global structure, we should be capable of exploiting it during intermediate updates and possibly also regularizing the results to be in the same low-rank space. In particular, at each iteration, instead of every $(s, a)$ pair (i.e., Eq. (1)), we would like to only calculate $Q^{(t+1)}$ for *some* $(s, a)$ pairs and then *exploit* the low-rank structure to recover the whole $Q^{(t+1)}$ matrix. We choose matrix estimation (ME) as our reconstruction oracle. The reconstructed matrix is often with low rank, and hence *regularizing* the $Q$ matrix to be low-rank as well. We validate this framework in Fig. 1(c) and 1(d), where for each iteration, we only randomly sample 50% of the $(s, a)$ pairs, calculate their corresponding $Q^{(t+1)}$ and reconstruct the whole $Q^{(t+1)}$ matrix with ME. Clearly, around 40 iterations, we obtain comparable results to the vanilla value iteration. Importantly, this comparable performance only needs to directly compute 50% of the whole $Q$ matrix at each iteration. It is not hard to see that in general, each vanilla value iteration incurs a computation cost of $O(|\mathcal{S}|^2|\mathcal{A}|^2)$. The complexity of our method however only scales as $O(p|\mathcal{S}|^2|\mathcal{A}|^2) + O_{ME}$, where $p$ is the percentage of pairs we sample and $O_{ME}$ is the complexity of ME. In general, many ME methods employ SVD as a subroutine, whose complexity is bounded by $O(\min\{|\mathcal{S}|^2|\mathcal{A}|, |\mathcal{S}||\mathcal{A}|^2\})$ (Trefethen & David Bau, 1997). For low-rank matrices, faster methods can have a complexity of order linear in the dimensions (Mazumder et al., 2010). In other words, our approach improves computational efficiency, especially for modern high-dimensional applications. This overall framework thus appears to be a successful technique: it exploits the low-rank behavior effectively and efficiently when the underlying task indeed possesses such a structure.

## 3 STRUCTURED VALUE-BASED PLANNING

Having developed the intuition underlying our methodology, we next provide a formal description in Sections 3.1 and 3.2. One natural question is whether such structures and our method are general in more realistic control tasks. Towards this end, we provide further empirical support in Section 3.3.

### 3.1 MATRIX ESTIMATION

ME considers about recovering a full data matrix, based on potentially incomplete and noisy observations of its elements. Formally, consider an unknown data matrix $X \in \mathbb{R}^{n \times m}$ and a set of observed entries $\Omega$. If the observations are incomplete, it is often assumed that each entry of $X$ is observed independently with probability $p \in (0, 1]$. In addition, the observation could be noisy, where the noise is assumed to be mean zero. Given such an observed set $\Omega$, the goal of ME is to produce an estimator $\hat{M}$ so that $||\hat{M} - X|| \approx 0$, under an appropriate matrix norm such as the Frobenius norm.

The algorithms in this field are rich. Theoretically, the essential message is: exact or approximate recovery of the data matrix $X$ is guaranteed if $X$ contains some global structure (Candès & Recht, 2009; Chatterjee et al., 2015; Chen & Chi, 2018). In the literature, most attention has been focusing on the *low-rank* structure of a matrix. Correspondingly, there are many provable, practical algorithms to achieve the desired recovery. Early convex optimization methods (Candès & Recht, 2009) seek to minimize the nuclear norm, $||\hat{M}||_*$, of the estimator. For example, fast algorithms, such as the Soft-Impute algorithm (Mazumder et al., 2010) solves the following minimization problem:

$$\min_{\hat{M} \in \mathbb{R}^{n \times m}} \frac{1}{2} \sum_{(i,j) \in \Omega} \left( \hat{M}_{ij} - X_{ij} \right)^2 + \lambda ||\hat{M}||_*. \tag{2}$$

Since the nuclear norm $|| \cdot ||_*$ is a convex relaxation of the rank, the convex optimization approaches favor solutions that are with small reconstruction errors and in the meantime being relatively low-rank, which are desirable for our applications. Apart from convex optimization, there are also spectral methods and even non-convex optimization approaches (Chatterjee et al., 2015; Chen & Wainwright, 2015; Ge et al., 2016). In this paper, we view ME as a principled reconstruction oracle to effectively exploit the low-rank structure. For faster computation, we mainly employ the Soft-Impute algorithm.

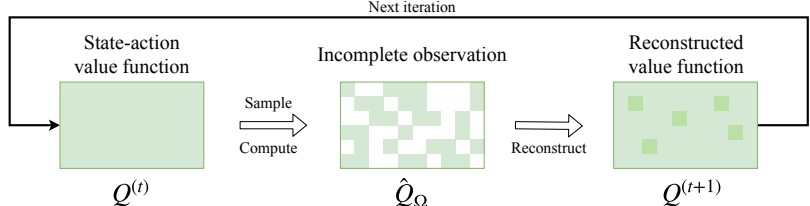

Figure 2: An illustration of the proposed SVP algorithm for leveraging low-rank structures.

## 3.2 OUR APPROACH: STRUCTURED VALUE-BASED PLANNING

We now formally describe our approach, which we refer as structured value-based planning (SVP). Fig. 2 illustrates our overall approach for solving MDP with a known model. The approach is based on the $Q$-value iteration. At the $t$-th iteration, instead of a full pass over all state-action pairs:

1. SVP first randomly selects a subset $\Omega$ of the state-action pairs. In particular, each state-action pair in $\mathcal{S} \times \mathcal{A}$ is observed (i.e., included in $\Omega$) independently with probability $p$.

2. For each selected $(s, a)$, the intermediate $\hat{Q}(s, a)$ is computed based on the $Q$-value iteration:

$$\hat{Q}(s,a) \leftarrow \sum_{s'} P(s'|s,a) \left( r(s,a) + \gamma \max_{a'} Q^{(t)}(s', a') \right), \quad \forall (s,a) \in \Omega.$$

3. The current iteration then ends by reconstructing the full $Q$ matrix with matrix estimation, from the set of observations in $\Omega$. That is, $Q^{(t+1)} = \text{ME}\big(\{\hat{Q}(s,a)\}_{(s,a)\in\Omega}\big)$.

Overall, each iteration reduces the computation cost by roughly $1 - p$ (cf. discussions in Section 2). In Appendix A, we provide the pseudo-code and additionally, a short discussion on the technical difficulty for theoretical analysis. Nevertheless, we believe that the consistent empirical benefits, as will be demonstrated, offer a sound foundation for future analysis.

## 3.3 EMPIRICAL EVALUATION ON STOCHASTIC CONTROL TASKS

We empirically evaluate our approach on several classical stochastic control tasks, including the Inverted Pendulum, the Mountain Car, the Double Integrator, and the Cart-Pole. Our objective is to demonstrate, as in the toy example, that if the optimal $Q^*$ has a low-rank structure, then the proposed SVP algorithm should be able to exploit the structure for efficient planning. We present the evaluation on Inverted Pendulum, and leave additional results on other planning tasks in Appendix B and C.

**Inverted Pendulum** In this classical continuous task, our goal is to balance an inverted pendulum to its upright position, and the performance metric is the average angular deviation. The dynamics is described by the angle and the angular speed, i.e., $s = (\theta, \dot{\theta})$, and the action $a$ is the torque applied. We discretize the task to have 2500 states and 1000 actions, leading to a $2500 \times 1000$ $Q$-value matrix. For different discretization scales, we provide further results in Appendix G.1.

**The Low-rank Structure** We first verify that the optimal $Q^*$ indeed contains the desired low-rank structure. We run the vanilla value iteration until it converges. The converged $Q$ matrix is found to have an approximate rank of 7. For further evidence, in Appendix C, we construct "low-rank" policies directly from the converged $Q$ matrix, and show that the policies maintain the desired performance.

**The SVP Policy** Having verified the structure, we would expect our approach to be effective. To this end, we apply SVP with different observation probability $p$ and fix the overall number of iterations to be the same as the vanilla $Q$-value iteration. Fig. 3 confirms the success of our approach. Fig. 3(a), 3(b) and 3(c) show the comparison between optimal policy and the final policy based on SVP. We further illustrate the performance metric, the average angular deviation, in Fig. 3(d). Overall, much fewer samples are needed for SVP to achieve a comparable performance to the optimal one.

## 4 STRUCTURED VALUE-BASED DEEP REINFORCEMENT LEARNING

So far, our focus has been on tabular MDPs where value iteration can be applied straightforwardly. However, the idea of exploiting structure is much more powerful: we propose a natural extension of

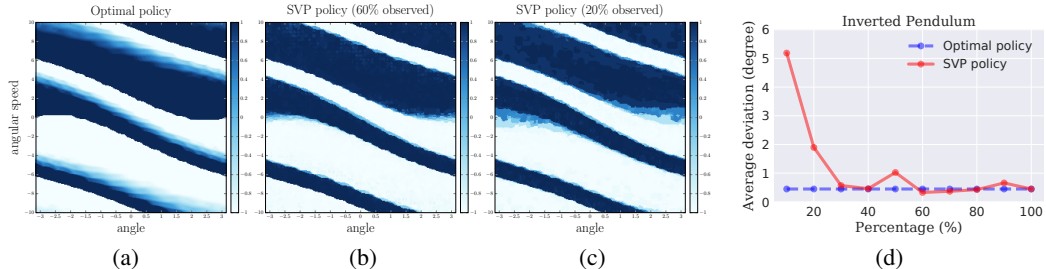

Figure 3: Performance comparison between optimal policy and the proposed SVP policy.

our approach to deep RL. Our scheme again intends to exploit and regularize structures in the $Q$-value function with ME. As such, it can be seamlessly incorporated into value-based RL techniques that include a $Q$-network. We demonstrate this on Atari games, across various value-based RL techniques.

### 4.1 EVIDENCE OF STRUCTURED $Q$-VALUE FUNCTION

Before diving into deep RL, let us step back and review the process we took to develop our intuition. Previously, we start by treating the $Q$-value as a matrix. To exploit the structure, we first verify that certain MDPs have essentially a low-rank $Q^*$. We argue that if this is indeed the case, then enforcing the low-rank structures throughout the iterations, by leveraging ME, should lead to better algorithms.

A naive extension of the above reasoning to deep RL immediately fails. In particular, with images as states, the state space is effectively infinitely large, leading to a tall $Q$ matrix with numerous number of rows (states). Verifying the low-rank structure for deep RL hence seems intractable. However, by definition, if a large matrix is low-rank, then almost any row is a linear combination of some other rows. That is, if we sample a small batch of the rows, the resulting matrix is most likely low-rank as well. To probe the structure of the deep $Q$ function, it is, therefore, natural to understand the rank of a randomly sampled batch of states. In deep RL, our target for exploring structures is no longer the optimal $Q^*$, which is never available. In fact, like SVP, the natural objective should be the converged values of the underlying algorithm, which in "deep" scenarios, are the eventually learned $Q$ function.

With the above discussions, we now provide evidence for the low-rank structure of learned $Q$ function on some Atari games. We train standard DQN on 4 games, with a batch size of 32. To be consistent, the 4 games all have 18 actions. After the training process, we randomly sample a batch of 32 states, evaluate with the learned $Q$ network and finally synthesize them into a matrix. That is, a $32 \times 18$ data matrix with rows the batch of states, the columns the actions, and the entries the values from the learned $Q$ network. Note that the rank of such a matrix is at most 18. The above process is repeated for 10,000 times, and the histogram and empirical CDF of the approximate rank is plotted in Fig. 4. Apparently, there is a strong evidence supporting a highly structured low-rank $Q$ function for those games – the approximate ranks are uniformly small; in most cases, they are around or smaller than 3.

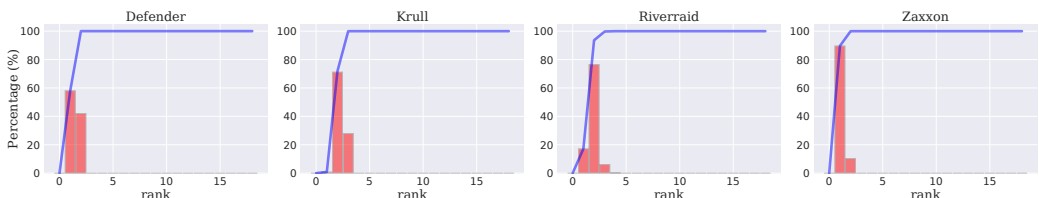

Figure 4: Approximate rank of different Atari games: histogram (red) and empirical CDF (blue).

### 4.2 OUR APPROACH: STRUCTURED VALUE-BASED RL

Having demonstrated the low-rank structure within some deep RL tasks, we naturally seek approaches that exploit the structure during the training process. We extend the same intuitions here: if eventually, the learned $Q$ function is of low rank, then enforcing/regularizing the low rank structure for each iteration of the learning process should similarly lead to efficient learning and potentially better performance. In deep RL, each iteration of the learning process is naturally the SGD step where one

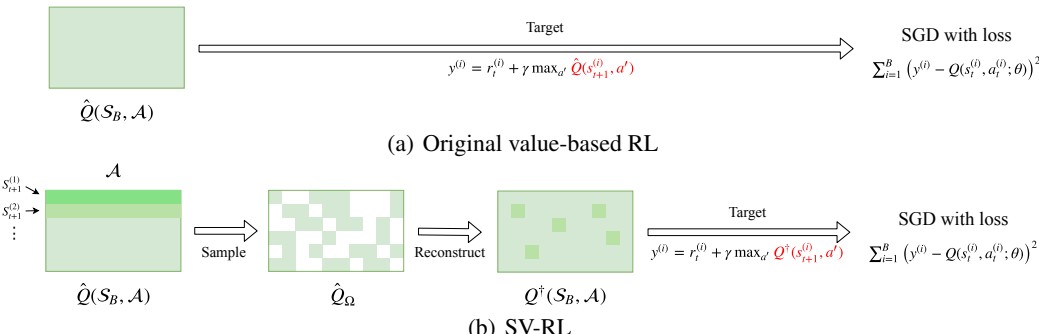

Figure 5: An illustration of the proposed SV-RL scheme, compared to the original value-based RL.

would update the $Q$ network. Correspondingly, this suggests us to harness the structure within the batch of states. Following our previous success, we leverage ME to achieve this task.

We now formally describe our approach, referred as structured value-based RL (SV-RL). It exploits the structure for the sampled batch at each SGD step, and can be easily incorporated into any $Q$-value based RL methods that update the $Q$ network via a similar step as in $Q$-learning. In particular, $Q$-value based methods have a common model update step via SGD, and we only exploit structure of the sampled batch at this step – the other details pertained to each specific method are left intact.

Precisely, when updating the model via SGD, $Q$-value based methods first sample a batch of $B$ transitions, $\{(s_t^{(i)}, r_t^{(i)}, a_t^{(i)}, s_{t+1}^{(i)})\}_{i=1}^B$, and form the following updating targets:

$$y^{(i)} = r_t^{(i)} + \gamma \max_{a'} \hat{Q}(s_{t+1}^{(i)}, a'). \tag{3}$$

For example, in DQN, $\hat{Q}$ is the target network. The $Q$ network is then updated by taking a gradient step for the loss function $\sum_{i=1}^B \left(y^{(i)} - Q(s_t^{(i)}, a_t^{(i)}; \theta)\right)^2$, with respect to the parameter $\theta$.

To exploit the structure, we then consider reconstructing a matrix $Q^\dagger$ from $\hat{Q}$, via ME. The reconstructed $Q^\dagger$ will replace the role of $\hat{Q}$ in Eq. (3) to form the targets $y^{(i)}$ for the gradient step. In particular, the matrix $Q^\dagger$ has a dimension of $B \times |\mathcal{A}|$, where the rows represent the "next states" $\{s_{t+1}^{(i)}\}_{i=1}^B$ in the batch, the columns represent actions, and the entries are reconstructed values. Let $\mathcal{S}_B \triangleq \{s_{t+1}^{(i)}\}_{i=1}^B$. The SV-RL alters the SGD update step as illustrated in Algorithm 1 and Fig. 5.

---

**Algorithm 1:** Structured Value-based RL (SV-RL)

1: follow the chosen value-based RL method (e.g., DQN) as usual.
2: **except** that for model updates with gradient descent, **do**
3:     /* exploit structure via matrix estimation*/
4:     sample a set $\Omega$ of state-action pairs from $\mathcal{S}_B \times \mathcal{A}$. In particular, each state-action pair in $\mathcal{S}_B \times \mathcal{A}$ is observed (i.e., included in $\Omega$) with probability $p$, independently.
5:     evaluate every state-action pair in $\Omega$ via $\hat{Q}$, where $\hat{Q}$ is the network that would be used to form the targets $\{y^{(i)}\}_{i=1}^B$ in the original value-based methods (cf. Eq. (3)).
6:     based on the evaluated values, reconstruct a matrix $Q^\dagger$ with ME, i.e.,

$$Q^\dagger = \text{ME}\big(\{\hat{Q}(s,a)\}_{(s,a) \in \Omega}\big).$$

7:     /* new targets with reconstructed $Q^\dagger$ for the gradient step*/
8:     replace $\hat{Q}$ in Eq. (3) with $Q^\dagger$ to evaluate the targets $\{y^{(i)}\}_{i=1}^B$, i.e.,

$$\text{SV-RL Targets:} \quad y^{(i)} = r_t^{(i)} + \gamma \max_{a'} Q^\dagger(s_{t+1}^{(i)}, a'). \tag{4}$$

9:     update the $Q$ network with the original targets replaced by the SV-RL targets.

---

Note the resemblance of the above procedure to that of SVP in Section 3.2. When the full $Q$ matrix is available, in Section 3.2, we sub-sample the $Q$ matrix and then reconstruct the entire matrix. When only a subset of the states (i.e., the batch) is available, naturally, we look at the corresponding sub-matrix of the entire $Q$ matrix, and seek to exploit its structure.

### 4.3 EMPIRICAL EVALUATION WITH VARIOUS VALUE-BASED METHODS

**Experimental Setup** We conduct extensive experiments on Atari 2600. We apply SV-RL on three representative value-based RL techniques, i.e., DQN, double DQN and dueling DQN. We fix the total number of training iterations and set all the hyper-parameters to be the same. For each experiment, averaged results across multiple runs are reported. Additional details are provided in Appendix D.

**Consistent Benefits for "Structured" Games** We present representative results of SV-RL applied to the three value-based deep RL techniques in Fig. 6. These games are verified by method mentioned in Section 4.1 to be low-rank. Additional results on all Atari games are provided in Appendix E. The figure reveals the following results. First, games that possess structures indeed benefit from our approach, earning mean rewards that are strictly higher than the vanilla algorithms across time. More importantly, we observe consistent improvements across different value-based RL techniques. This highlights the important role of the intrinsic structures, which are independent of the specific techniques, and justifies the effectiveness of our approach in consistently exploiting such structures.

**Further Observations** Interestingly however, the performance gains vary from games to games. Specifically, the majority of the games can have benefits, with few games performing similarly or slightly worse. Such observation motivates us to further diagnose SV-RL in the next section.

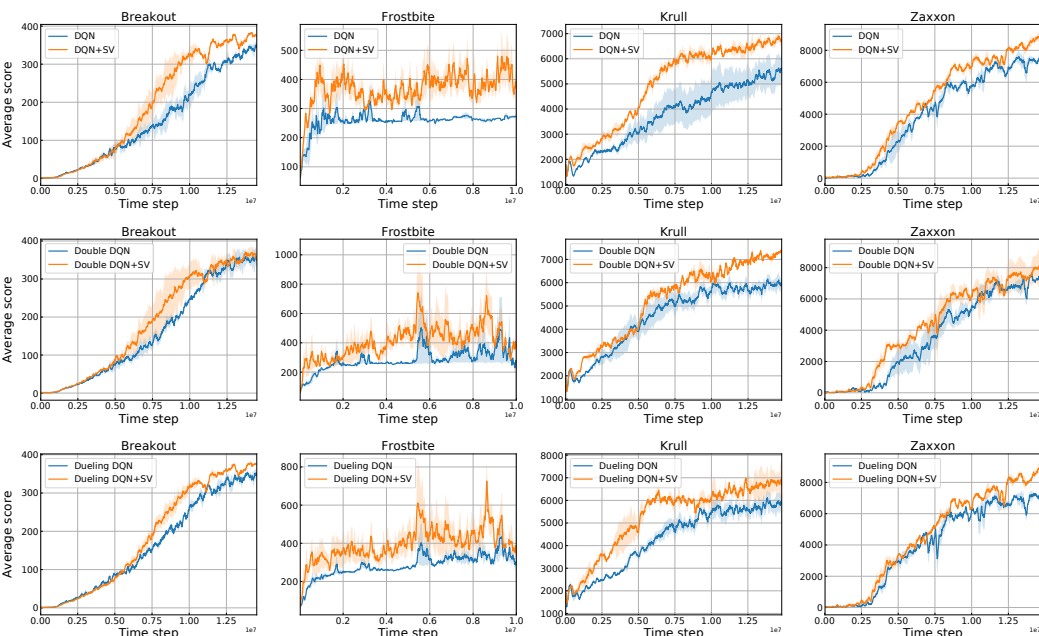

Figure 6: Results of SV-RL on various value-based deep RL techniques. **First row:** results on DQN. **Second row:** results on double DQN. **Third row:** results on dueling DQN.

## 5 DIAGNOSE AND INTERPRET PERFORMANCE IN DEEP RL

So far, we have demonstrated that games which possess structured $Q$-value functions can consistently benefit from SV-RL. Obviously however, not all tasks in deep RL would possess such structures. As such, we seek to diagnose and further interpret our approach at scale.

**Diagnosis** We select 4 representative examples (with 18 actions) from all tested games, in which SV-RL performs better on two tasks (i.e., FROSTBITE and KRULL), slightly better on one task (i.e., ALIEN), and slightly worse on the other (i.e., SEAQUEST). The intuitions we developed in Section 4

incentivize us to further check the approximate rank of each game. As shown in Fig. 7, in the two better cases, both games are verified to be approximately low-rank ($\sim 2$), while the approximate rank in ALIEN is moderately high ($\sim 5$), and even higher in SEAQUEST ($\sim 10$).

**Consistent Interpretations** As our approach is designed to exploit structures, we would expect to attribute the differences in performance across games to the "strength" of their structured properties. Games with strong low-rank structures tend to have larger improvements with SV-RL (Fig. 7(a) and 7(b)), while moderate approximate rank tends to induce small improvements (Fig. 7(c)), and high approximate rank may induce similar or slightly worse performances (Fig. 7(d)). The empirical results are well aligned with our arguments: if the $Q$-function for the task contains low-rank structure, SV-RL can exploit such structure for better efficiency and performance; if not, SV-RL may introduce slight or no improvements over the vanilla algorithms. As mentioned, the ME solutions balance being low rank and having small reconstruction error, which helps to ensure a reasonable or only slightly degraded performance, even for "high rank" games. We further observe consistent results on ranks *vs.* improvement across different games and RL techniques in Appendix E, verifying our arguments.

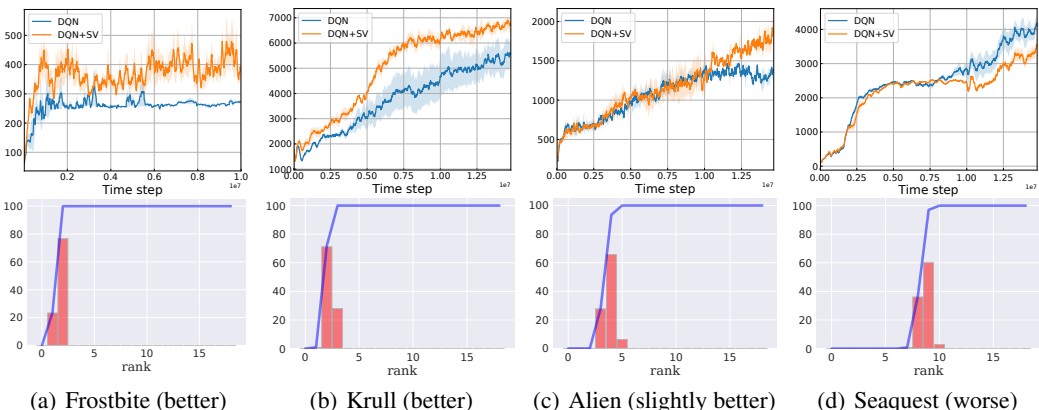

(a) Frostbite (better)  (b) Krull (better)  (c) Alien (slightly better)  (d) Seaquest (worse)

Figure 7: Interpretation of deep RL results. We plot games where the SV-based method performs differently. More structured games (with lower rank) can achieve better performance with SV-RL.

## 6  RELATED WORK

**Structures in Value Function** Recent work in the control community starts to explore the structure of value function in control/planning tasks (Ong, 2015; Alora et al., 2016; Gorodetsky et al., 2015; 2018). These work focuses on decomposing the value function and subsequently operating on the reduced-order space. In spirit, we also explore the low-rank structure in value function. The central difference is that instead of decomposition, we focus on "completion". We seek to efficiently operate on the original space by looking at few elements and then leveraging the structure to infer the rest, which allows us to extend our approach to modern deep RL. In addition, while there are few attempts for basis function learning in high dimensional RL (Liang et al., 2016), functions are hard to generate in many cases and approaches based on basis functions typically do not get the same performance as DQN, and do not generalize well. In contrast, we provide a principled and systematic method, which can be applied to any framework that employs value-based methods or sub-modules. Finally, we remark that at a higher level, there are studies exploiting different structures within a task to design effective algorithms, such as exploring the low-dimensional system dynamics in predictive state representation (Singh et al., 2012) or exploring the so called low "Bellman rank" property (Jiang et al., 2017). We provide additional discussions in Appendix F.

**Value-based Deep RL** Deep RL has emerged as a promising technique, highlighted by key successes such as solving Atari games via $Q$-learning (Mnih et al., 2015) and combining with Monte Carlo tree search (Browne et al., 2012; Shah et al., 2019) to master Go, Chess and Shogi (Silver et al., 2017a;b). Methods based on learning value functions are fundamental components in deep RL, exemplified by the deep $Q$-network (Mnih et al., 2013; 2015). Over the years, there has been a large body of literature on its variants, such as double DQN (Van Hasselt et al., 2016), dueling DQN (Wang et al., 2015), IQN (Dabney et al., 2018) and other techniques that aim to improve exploration (Osband et al., 2016; Ostrovski et al., 2017). Our approach focuses on general value-based RL methods, rather than specific algorithms. As long as the method has a similar model update step as in $Q$-learning, our

approach can leverage the structure to help with the task. We empirically demonstrate that deep RL tasks that have structured value functions indeed benefit from our scheme.

**Matrix Estimation** ME is the primary tool we leverage to exploit the low-rank structure in value functions. Early work in the field is primarily motivated by recommendation systems. Since then, the techniques have been widely studied and applied to different domains (Abbe & Sandon, 2015; Borgs et al., 2017), and recently even in robust deep learning (Yang et al., 2019). The field is relatively mature, with extensive algorithms and provable recovery guarantees for structured matrix (Davenport & Romberg, 2016; Chen & Chi, 2018). Because of the strong promise, we view ME as a principled reconstruction oracle to exploit the low-rank structures within matrices.

## 7 CONCLUSION

We investigated the structures in value function, and proposed a complete framework to understand, validate, and leverage such structures in various tasks, from planning to deep reinforcement learning. The proposed SVP and SV-RL algorithms harness the strong low-rank structures in the $Q$ function, showing consistent benefits for both planning tasks and value-based deep reinforcement learning techniques. Extensive experiments validated the significance of the proposed schemes, which can be easily embedded into existing planning and RL frameworks for further improvements.

### ACKNOWLEDGMENTS

The authors would like to thank Luxin Zhang, the members of NETMIT and CSAIL, and the anonymous reviewers for their insightful comments and helpful discussion. Zhi Xu is supported by the Siemens FutureMakers Fellowship.

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

# A  PSEUDO CODE AND DISCUSSIONS FOR STRUCTURED VALUE-BASED PLANNING (SVP)

---

**Algorithm 2:** Structured Value-based Planning (SVP)

---

1: **Input:** initialized value function $Q^{(0)}(s,a)$ and prescribed observing probability $p$.
2: **for** $t = 1, 2, 3, \ldots$ **do**
3:     randomly sample a set $\Omega$ of observed entries from $\mathcal{S} \times \mathcal{A}$, each with probability $p$
4:     `/* update the randomly selected state-action pairs*/`
5:     **for** each state-action pair $(s,a) \in \Omega$ **do**
6:

$$\hat{Q}(s,a) \leftarrow \sum_{s'} P(s'|s,a) \left( r(s,a) + \gamma \max_{a'} Q^{(t)}(s',a') \right)$$

7:     **end for**
8:     `/* reconstruct the Q matrix via matrix estimation*/`
9:     apply matrix completion to the observed values $\{\hat{Q}(s,a)\}_{(s,a)\in\Omega}$ to reconstruct $Q^{(t+1)}$:

$$Q^{(t+1)} \leftarrow \mathrm{ME}\big(\{\hat{Q}(s,a)\}_{(s,a)\in\Omega}\big)$$

10: **end for**

---

While based on classical value iteration, we remark that a theoretical analysis, even in the tabular case, is quite complex. (1) Although the field of ME is somewhat mature, the analysis has been largely focused on the "one-shot" problem: recover a static data matrix given one incomplete observation. Under the iterative scenario considered here, standard assumptions are easily broken and the analysis warrants potentially new machinery. (2) Furthermore, much of the effort in the ME community has been devoted to the Frobenius norm guarantees rather than the infinite norm as in value iteration. Non-trivial infinite norm bound has received less attention and often requires special techniques (Ding & Chen, 2018; Fan et al., 2016). Resolving the above burdens would be important future avenues in its own right for the ME community. Henceforth, this paper focuses on empirical analysis and more importantly, *generalizing* the framework successfully to modern deep RL contexts. As we will demonstrate, the consistent empirical benefits offer a sounding foundation for future analysis.

# B  EXPERIMENTAL SETUP OF STOCHASTIC CONTROL TASKS

**Inverted Pendulum**  As stated earlier in Sec. 3.3, the goal is to balance the inverted pendulum on the upright equilibrium position. The physical dynamics of the system is described by the angle and the angular speed, i.e., $(\theta, \dot{\theta})$. Denote $\tau$ as the time interval between decisions, $u$ as the torque input on the pendulum, the dynamics can be written as (Ong, 2015; Sutton & Barto, 2018):

$$\theta := \theta + \dot{\theta}\,\tau, \tag{5}$$

$$\dot{\theta} := \dot{\theta} + \left(\sin\theta - \dot{\theta} + u\right)\tau. \tag{6}$$

A reward function that penalizes control effort while favoring an upright pendulum is used:

$$r(\theta, u) = -0.1u^2 + \exp\left(\cos\theta - 1\right). \tag{7}$$

In the simulation, the state space is $(-\pi, \pi]$ for $\theta$ and $[-10, 10]$ for $\dot{\theta}$. We limit the input torque in $[-1, 1]$ and set $\tau = 0.3$. We discretize each dimension of the state space into 50 values, and action space into 1000 values, which forms an $Q$-value function a matrix of dimension $2500 \times 1000$. We follow (Julier & Uhlmann, 2004) to handle the policy of continuous states by modelling their transitions using multi-linear interpolation.

**Mountain Car**  We select another classical control problem, i.e., the Mountain Car (Sutton & Barto, 2018), for further evaluations. In this problem, an under-powered car aims to drive up a steep hill (Sutton & Barto, 2018). The physical dynamics of the system is described by the position and the

velocity, i.e., $(x, \dot{x})$. Denote $u$ as the acceleration input on the car, the dynamics can be written as

$$x := x + \dot{x}, \tag{8}$$

$$\dot{x} := \dot{x} - 0.0025 \cos(3x) + 0.001u. \tag{9}$$

The reward function is defined to encourage the car to get onto the top of the mountain at $x_0 = 0.5$:

$$r(x) = \begin{cases} 10, & x \geq x_0, \\ -1, & else. \end{cases} \tag{10}$$

We follow standard settings to restrict the state space as $[-0.07, 0.07]$ for $x$ and $[-1.2, 0.6]$ for $\dot{x}$, and limit the input $u \in [-1, 1]$. Similarly, the whole state space is discretized into 2500 values, and the action space is discretized into 1000 values. The evaluation metric we are concerned about is the total time it takes to reach the top of the mountain, given a randomly and uniformly generated initial state.

**Double Integrator**  We consider the Double Integrator system (Ren & Beard, 2008), as another classical control problem for evaluation. In this problem, a unit mass brick moving along the $x$-axis on a frictionless surface, with a control input which provides a horizontal force, $u$ (Tedrake, 2019). The task is to design a control system to regulate this brick to $\boldsymbol{x} = [0, 0]^T$. The physical dynamics of the system is described by the position and the velocity (i.e., $(x, \dot{x})$), and can be derived as

$$x := x + \dot{x}\,\tau, \tag{11}$$

$$\dot{x} := \dot{x} + u\,\tau. \tag{12}$$

Follow Tedrake (2019), we use the quadratic cost formulation to define the reward function, which regulates the brick to $\boldsymbol{x} = [0, 0]^T$:

$$r(x, \dot{x}) = -\frac{1}{2}\left(x^2 + \dot{x}^2\right). \tag{13}$$

We follow standard settings to restrict the state space as $[-3, 3]$ for both $x$ and $\dot{x}$, limit the input $u \in [-1, 1]$ and set $\tau = 0.1$. The whole state space is discretized into 2500 values, and the action space is discretized into 1000 values. Similarly, we define the evaluation metric as the total time it takes to reach to $\boldsymbol{x} = [0, 0]^T$, given a randomly and uniformly generated initial state.

**Cart-Pole**  Finally, we choose the Cart-Pole problem (Barto et al., 1983), a harder control problem with 4-dimensional state space. The problem consists a pole attached to a cart moving on a frictionless track. The cart can be controlled by means of a limited force within $10N$ that is possible to apply both to the left or to the right of the cart. The goal is to keep the pole on the upright equilibrium position. The physical dynamics of the system is described by the angle and the angular speed of the pole, and the position and the speed of the cart, i.e., $(\theta, \dot{\theta}, x, \dot{x})$. Denote $\tau$ as the time interval between decisions, $u$ as the force input on the cart, the dynamics can be written as

$$\ddot{\theta} := \frac{g \sin\theta - \frac{u + ml\dot{\theta}^2 \sin\theta}{m_c + m} \cos\theta}{l\left(\frac{4}{3} - \frac{m\cos^2\theta}{m_c + m}\right)}, \tag{14}$$

$$\ddot{x} := \frac{u + ml\left(\dot{\theta}^2 \sin\theta - \ddot{\theta}\cos\theta\right)}{m_c + m}, \tag{15}$$

$$\theta := \theta + \dot{\theta}\,\tau, \tag{16}$$

$$\dot{\theta} := \dot{\theta} + \ddot{\theta}\,\tau, \tag{17}$$

$$x := x + \dot{x}\,\tau, \tag{18}$$

$$\dot{x} := \dot{x} + \ddot{x}\,\tau, \tag{19}$$

where $g = 9.8m/s^2$ corresponds to the gravity acceleration, $m_c = 1kg$ denotes the mass of the cart, $m = 0.1kg$ denotes the mass of the pole, $l = 0.5m$ is half of the pole length, and $u$ corresponds to the force applied to the cart, which is limited by $u \in [-10, 10]$.

A reward function that favors the pole in an upright position, i.e., characterized by keeping the pole in vertical position between $|\theta| \leq \frac{12\pi}{180}$, is expressed as

$$r(\theta) = \cos^4(15\theta). \tag{20}$$

In the simulation, the state space is $[-\frac{\pi}{2}, \frac{\pi}{2}]$ for $\theta$, $[-3, 3]$ for $\dot{\theta}$, $[-2.4, 2.4]$ for $x$ and $[-3.5, 3.5]$ for $\dot{x}$. We limit the input force in $[-10, 10]$ and set $\tau = 0.1$. We discretize each dimension of the state space into 10 values, and action space into 1000 values, which forms an $Q$-value function a matrix of dimension $10000 \times 1000$.

## C  ADDITIONAL RESULTS FOR SVP

### C.1  INVERTED PENDULUM

We further verify that the optimal $Q^*$ indeed contains the desired low-rank structure. To this end, we construct "low-rank" policies directly from the converged $Q$ matrix. In particular, for the converged $Q$ matrix, we sub-sample a certain percentage of its entries, reconstruct the whole matrix via ME, and finally construct a corresponding policy. Fig. 8 illustrates the results, where the policy heatmap as well as the performance (i.e., the angular error) of the "low-rank" policy is essentially identical to the optimal one. The results reveal the intrinsic strong low-rank structures lie in the $Q$-value function.

We provide additional results for the inverted pendulum problem. We show the policy trajectory (i.e., how the angle of the pendulum changes with time) and the input changes (i.e., how the input torque changes with time), for each policy.

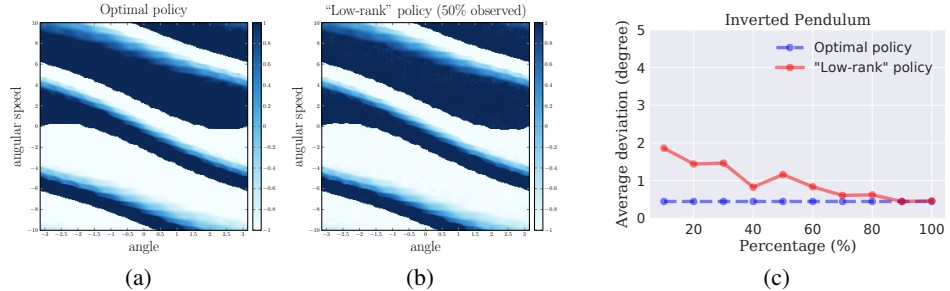

Figure 8: Performance comparison between optimal policy and the reconstructed "low-rank" policy.

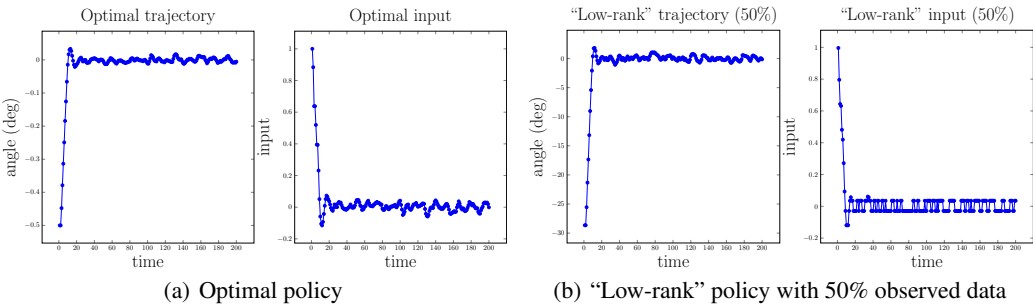

Figure 9: Comparison of the policy trajectories and the input torques between the two schemes.

In Fig. 9, we first show the comparison between the optimal policy and a "low-rank" policy. Recall that the low-rank policies are directly reconstructed from the converged $Q$ matrix, with limited observation of a certain percentage of the entries in the converged $Q$ matrix. As shown, the "low-rank" policy performs nearly identical to the optimal one, in terms of both the policy trajectory and the input torque changes. This again verifies the strong low-rank structure lies in the $Q$ function.

Further, we show the policy trajectory and the input torque changes of the SVP policy. We vary the percentage of observed data for SVP, and present the policies with $20\%$ and $60\%$ for demonstration. As reported in Fig. 10, the SVP policies are essentially identical to the optimal one. Interestingly, when we further decrease the observing percentage to $20\%$, the policy trajectory vibrates a little bit, but can still stabilize in the upright position with a small average angular deviation $\leq 5°$.

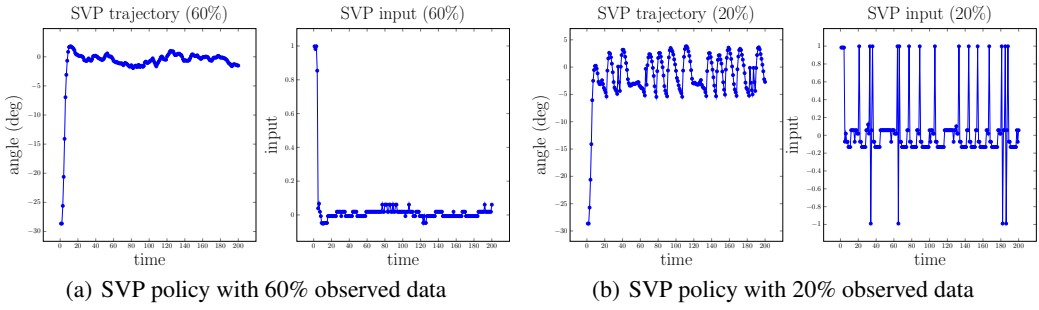

(a) SVP policy with 60% observed data      (b) SVP policy with 20% observed data

Figure 10: The policy trajectories and the input torques of the proposed SVP scheme.

## C.2   MOUNTAIN CAR

Similarly, we first verify the optimal $Q^*$ contains the desired low-rank structure. We use the same approach to generate a "low-rank" policy based on the converged optimal value function. Fig. 11(a) and 11(b) show the policy heatmaps, where the reconstructed "low-rank" policy maintains visually identical to the optimal one. In Fig. 11(c) and 12, we quantitatively show the average time-to-goal, the policy trajectory and the input changes between the two schemes. Compared to the optimal one, even with limited sampled data, the reconstructed policy can maintain almost identical performance.

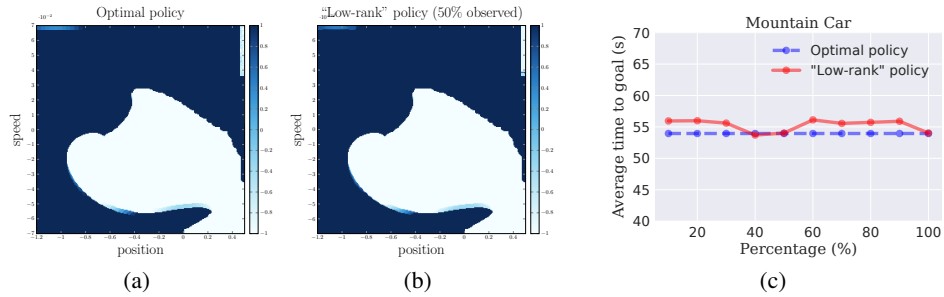

(a)        (b)        (c)

Figure 11: Performance comparison between optimal policy and the reconstructed "low-rank" policy.

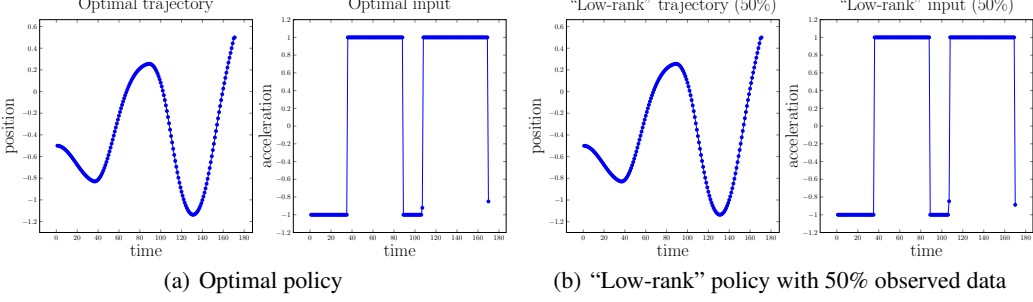

(a) Optimal policy      (b) "Low-rank" policy with 50% observed data

Figure 12: Comparison of the policy trajectories and the input changes between the two schemes.

We further show the results of the SVP policy with different amount of observed data (i.e., $20\%$ and $60\%$) in Fig. 13 and 14. Again, the SVP policies show consistently comparable results to the optimal policy, over various evaluation metrics. Interestingly, the converged $Q$ matrix of vanilla value iteration is found to have an approximate rank of $4$ (the whole matrix is $2500 \times 1000$), thus the SVP can harness such strong low-rank structure for perfect recovery even with only $20\%$ observed data.

## C.3   DOUBLE INTEGRATOR

For the Double Integrator, We first use the same approach to generate a "low-rank" policy. Fig. 15(a) and 15(b) show that the reconstructed "low-rank" policy is visually identical to the optimal one. In

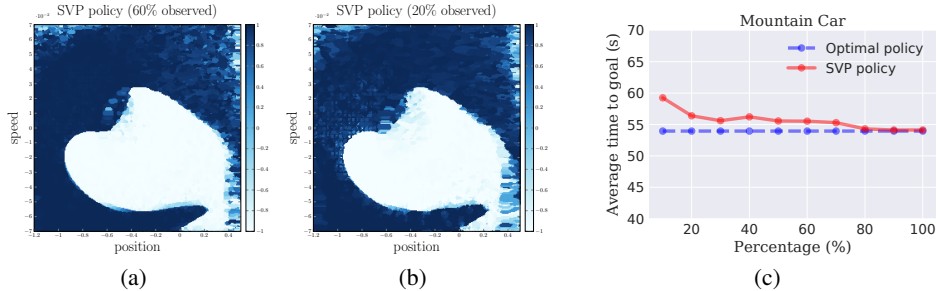

Figure 13: Performance of the proposed SVP policy, with different amount of observed data.

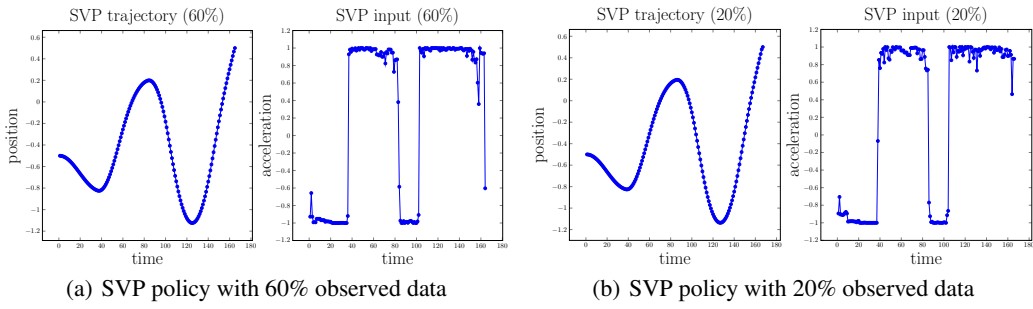

(a) SVP policy with 60% observed data   (b) SVP policy with 20% observed data

Figure 14: The policy trajectories and the input changes of the proposed SVP scheme.

Fig. 15(c) and 16, we quantitatively show the average time-to-goal, the policy trajectory and the input changes between the two schemes, where the reconstructed policy can achieve the same performance.

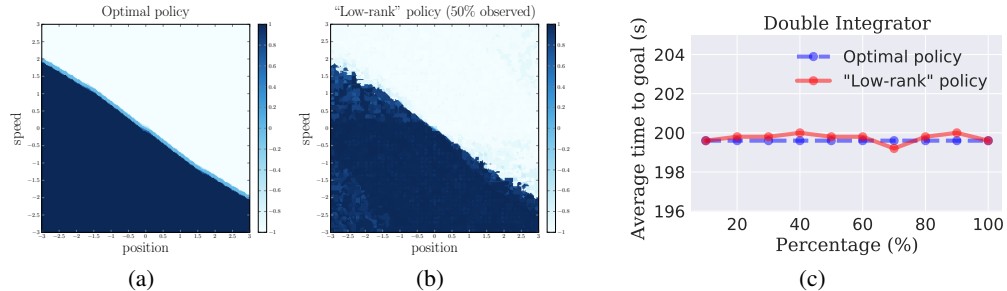

Figure 15: Performance comparison between optimal policy and the reconstructed "low-rank" policy.

Further, we show the results of the SVP policy with different amount of observed data (i.e., 20% and 60%) in Fig. 17 and 18. As shown, the SVP policies show consistently decent results, which demonstrates that SVP can harness such strong low-rank structure even with only 20% observed data.

## C.4 CART-POLE

Finally, we evaluate SVP on the Cart-Pole system. Note that since the state space has a dimension of 4, the policy heatmap should contain 4 dims, but is hard to visualize. Since the metric we care is the angle deviation, we here only plot the first two dims (i.e., the $(\theta, \dot{\theta})$ tuple) with fixed $x$ and $\dot{x}$, to visualize the policy heatmaps. We first use the same approach to generate a "low-rank" policy. Fig. 19(a) and 19(b) show the policy heatmaps, where the reconstructed "low-rank" policy is visually identical to the optimal one. In Fig. 19(c) and 20, we quantitatively show the average time-to-goal, the policy trajectory and the input changes between the two schemes. As demonstrated, the reconstructed policy can maintain almost identical performance with only small amount of sampled data.

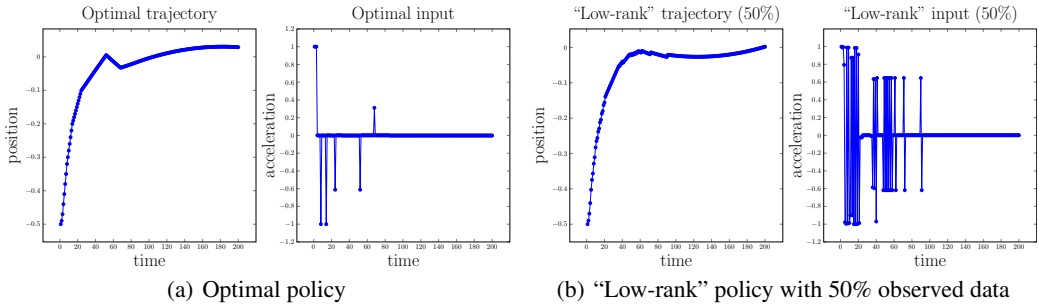

Figure 16: Comparison of the policy trajectories and the input changes between the two schemes.

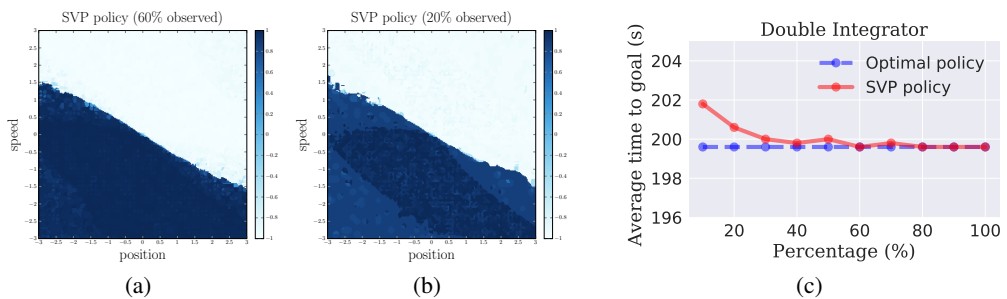

Figure 17: Performance of the proposed SVP policy, with different amount of observed data.

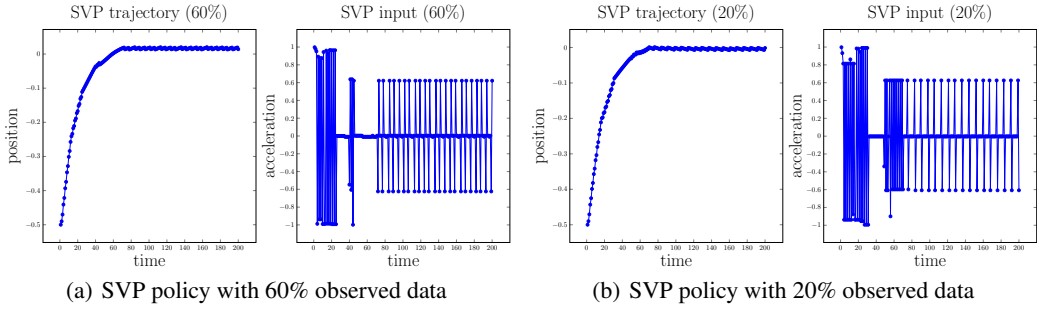

Figure 18: The policy trajectories and the input changes of the proposed SVP scheme.

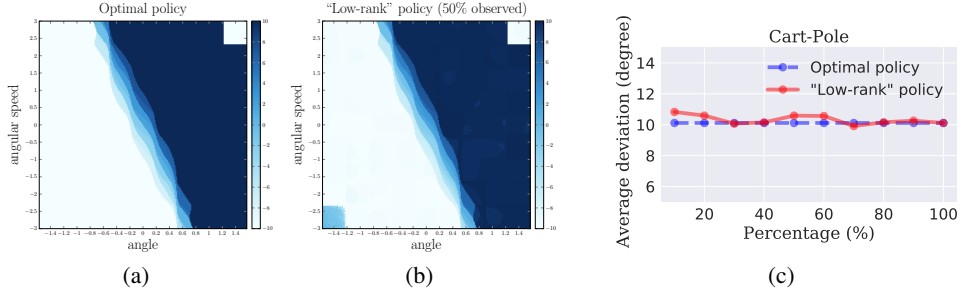

Figure 19: Performance comparison between optimal policy and the reconstructed "low-rank" policy.

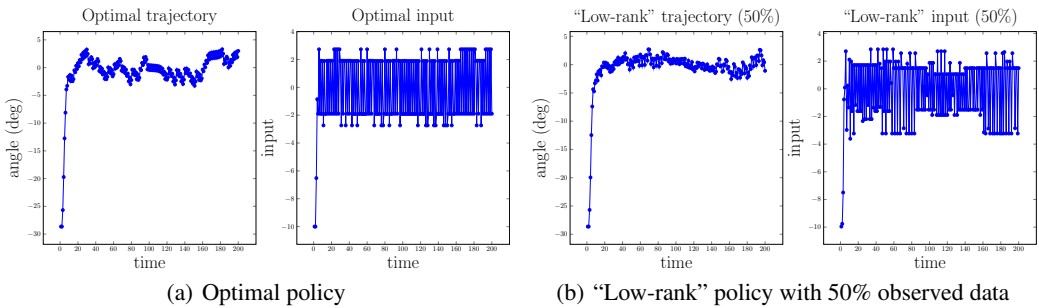

(a) Optimal policy

(b) "Low-rank" policy with 50% observed data

Figure 20: Comparison of the policy trajectories and the input changes between the two schemes.

We finally show the results of the SVP policy with different amount of observed data (i.e., 20% and 60%) in Fig. 21 and 22. Even for harder control tasks with higher dimensional state space, the SVP policies are still essentially identical to the optimal one. Across various stochastic control tasks, we demonstrate that SVP can consistently leverage strong low-rank structures for efficient planning.

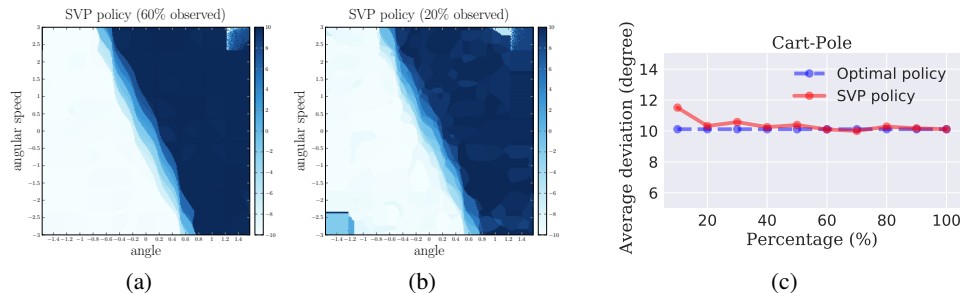

(a)             (b)             (c)

Figure 21: Performance of the proposed SVP policy, with different amount of observed data.

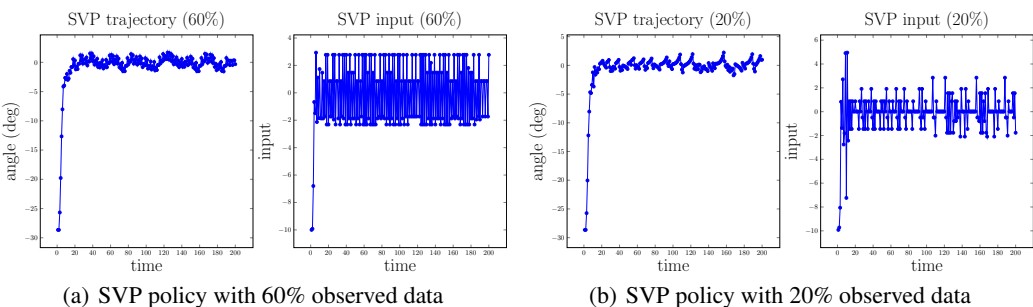

(a) SVP policy with 60% observed data

(b) SVP policy with 20% observed data

Figure 22: The policy trajectories and the input changes of the proposed SVP scheme.

# D    TRAINING DETAILS OF STRUCTURED VALUE-BASED RL (SV-RL)

**Training Details and Hyper-parameters**  The network architectures of DQN and dueling DQN used in our experiment are exactly the same as in the original papers (Mnih et al., 2013; Van Hasselt et al., 2016; Wang et al., 2015). We train the network using the Adam optimizer (Kingma & Ba, 2014). In all experiments, we set the hyper-parameters as follows: learning rate $\alpha = 1e^{-5}$, discount coefficient $\gamma = 0.99$, and a minibatch size of 32. The number of steps between target network updates is set to $10,000$. We use a simple exploration policy as the $\epsilon$-greedy policy with the $\epsilon$ decreasing linearly from 1 to 0.01 over $3e^5$ steps. For each experiment, we perform at least 3 independent runs and report the averaged results.

**SV-RL Details**  To reconstruct the matrix $Q^\dagger$ formed by the current batch of states, we mainly employ the Soft-Impute algorithm (Mazumder et al., 2010) throughout our experiments. We set the sub-sample rate to $p = 0.9$ of the $Q$ matrix, and use a linear scheduler to increase the sampling rate every $2e^6$ steps.

# E    ADDITIONAL RESULTS FOR SV-RL

**Experiments across Various Value-based RL**  We show more results across DQN, double DQN and dueling DQN in Fig. 23, 24, 25, 26, 27 and 28, respectively. For DQN, we complete **all 57 Atari games** using the proposed SV-RL, and verify that the majority of tasks contain low-rank structures (43 out of 57), where we can obtain consistent benefits from SV-RL. For each experiment, we associate the performance on the Atari game with its approximate rank. As mentioned in the main text, majority of the games benefit consistently from SV-RL. We note that roughly only **4** games, which have a significantly large rank, perform slightly worse than the vanilla DQN.

**Consistency and Interpretation**  Across all the experiments we have done, we observe that when the game possesses structures (i.e., being approximately low-rank), SV-RL can consistently improve the performance of various value-based RL techniques. The superior performance is maintained through most of the experiments, verifying the ability of the proposed SV-RL to harness the structures for better efficiency and performance in value-based deep RL tasks. In the meantime, when the approximate rank is relatively higher (e.g., SEAQUEST), the performance of SV-RL can be similar or worse than the vanilla scheme, which also aligns well with our intuitions. Note that the majority of the games have an action space of size 18 (i.e., rank is at most 18), while some (e.g., PONG) only have 6 or less (i.e., rank is at most 6).

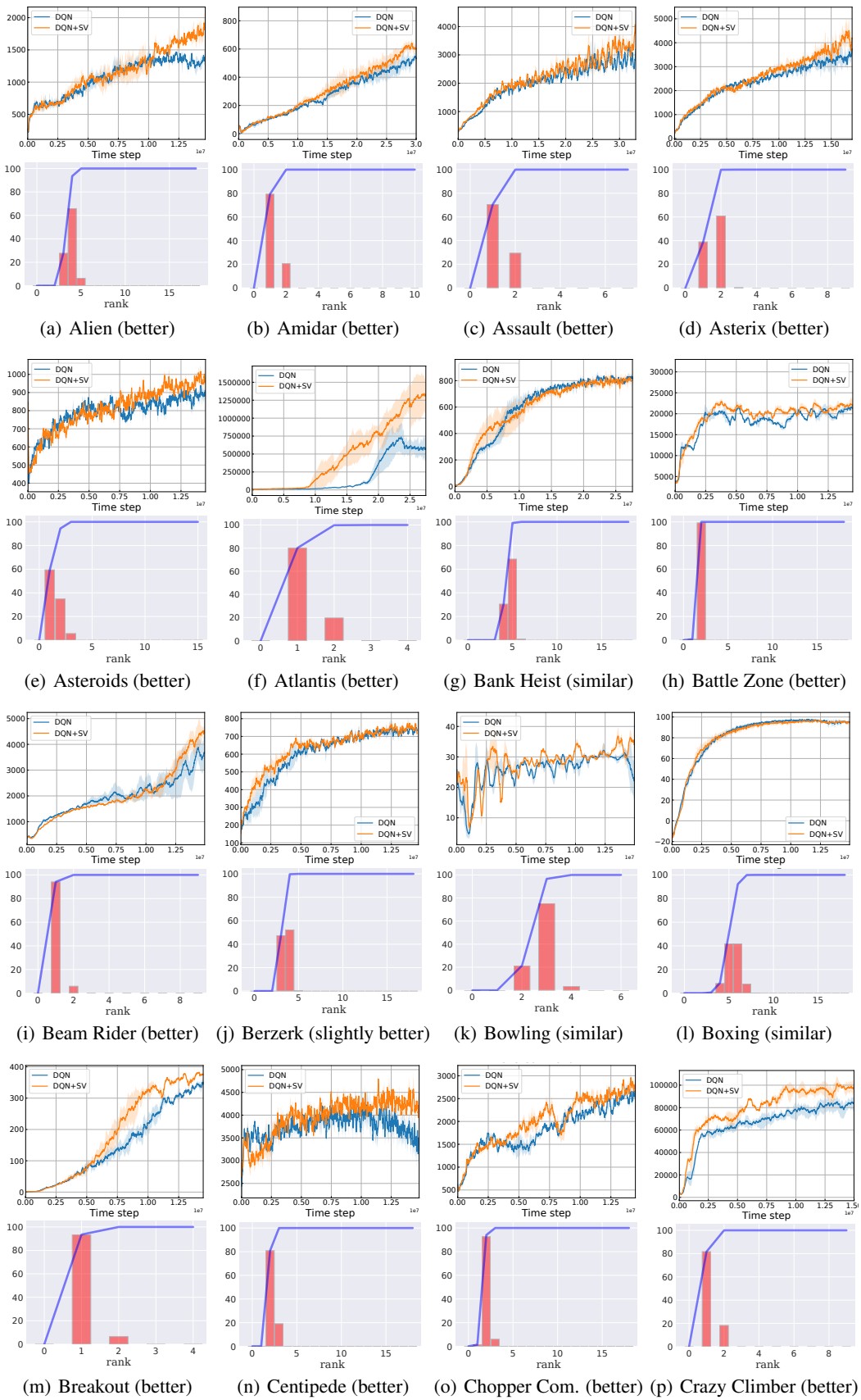

Figure 23: Additional results of SV-RL on DQN (Part A).

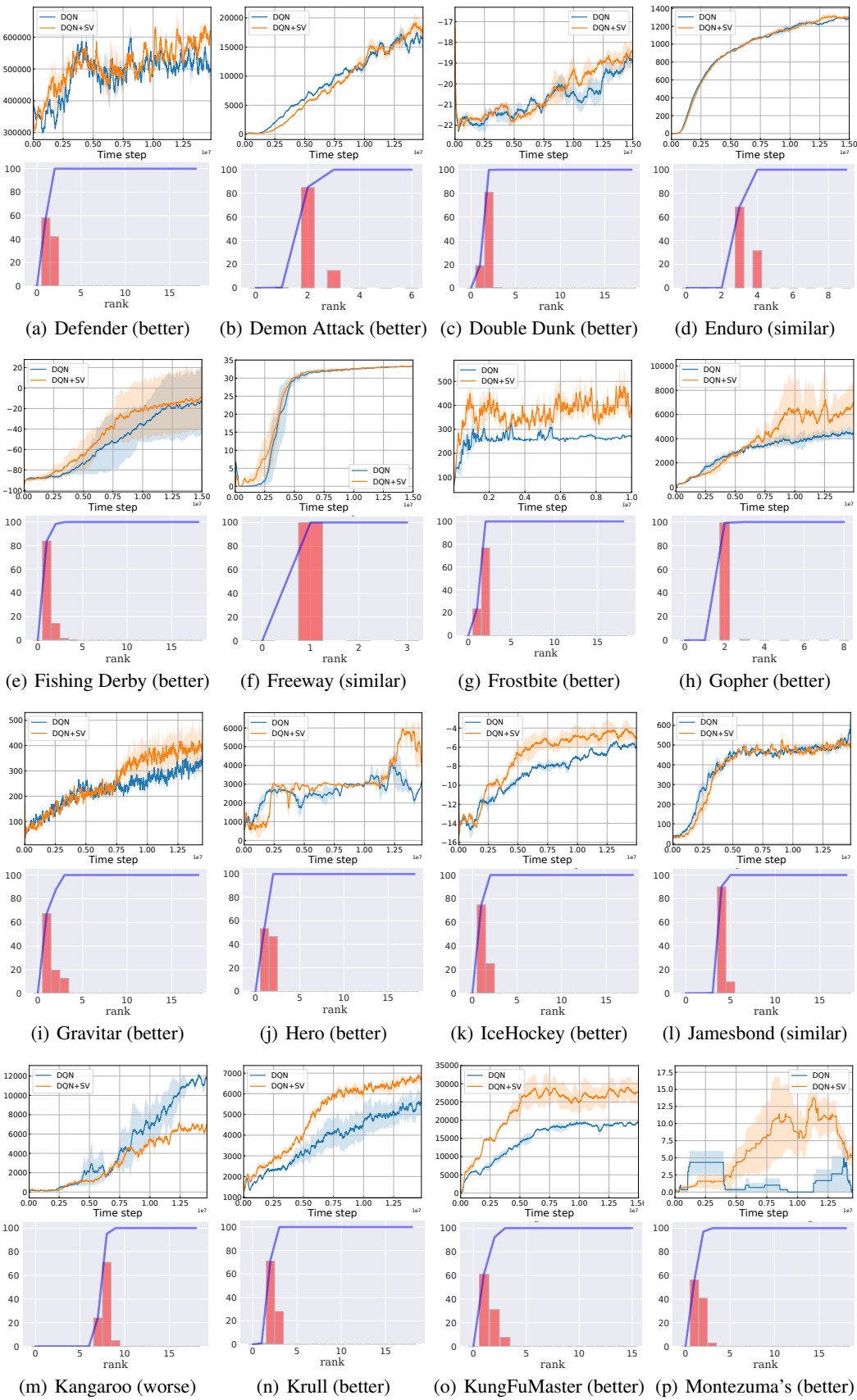

Figure 24: Additional results of SV-RL on DQN (Part B).

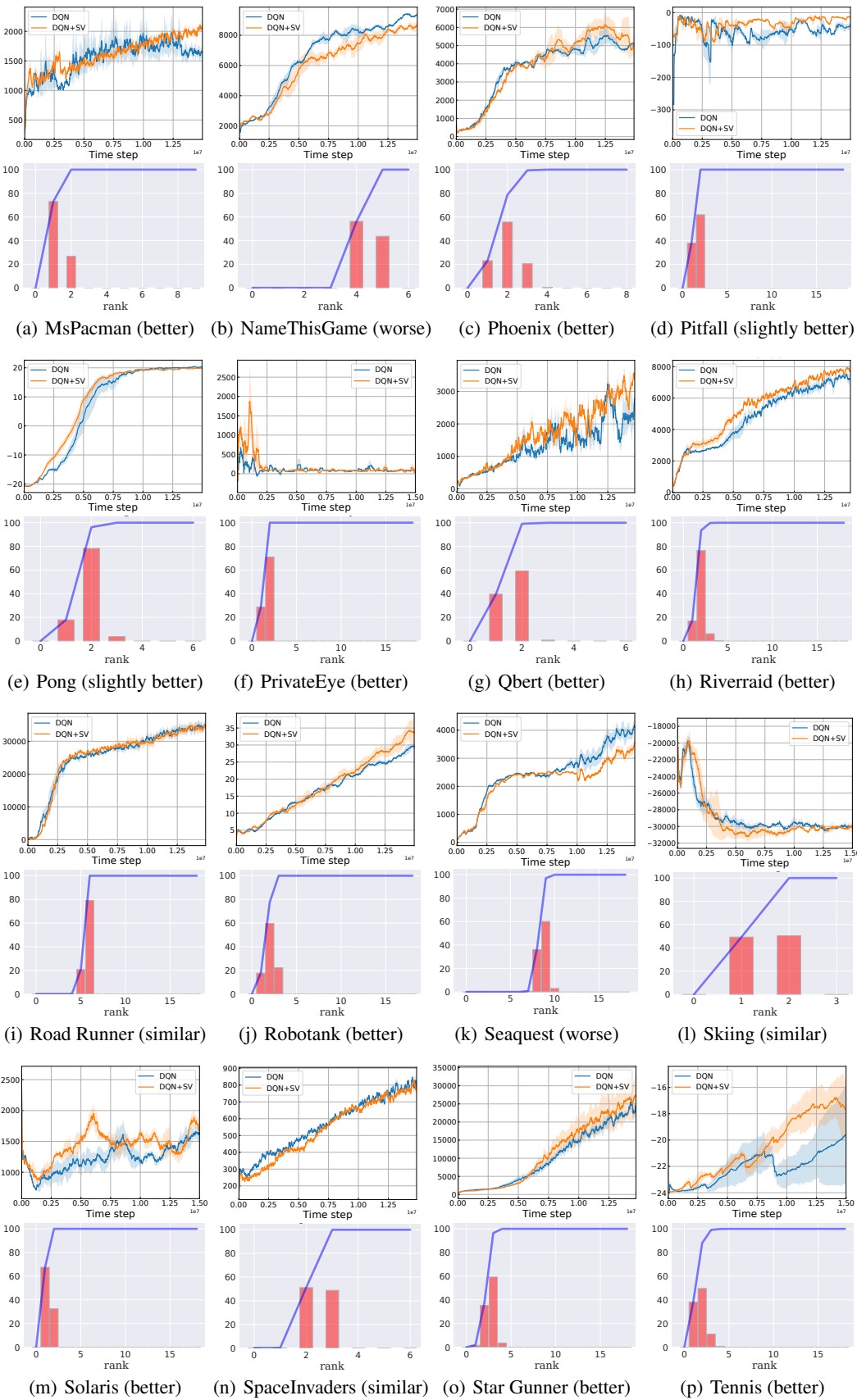

Figure 25: Additional results of SV-RL on DQN (Part C).

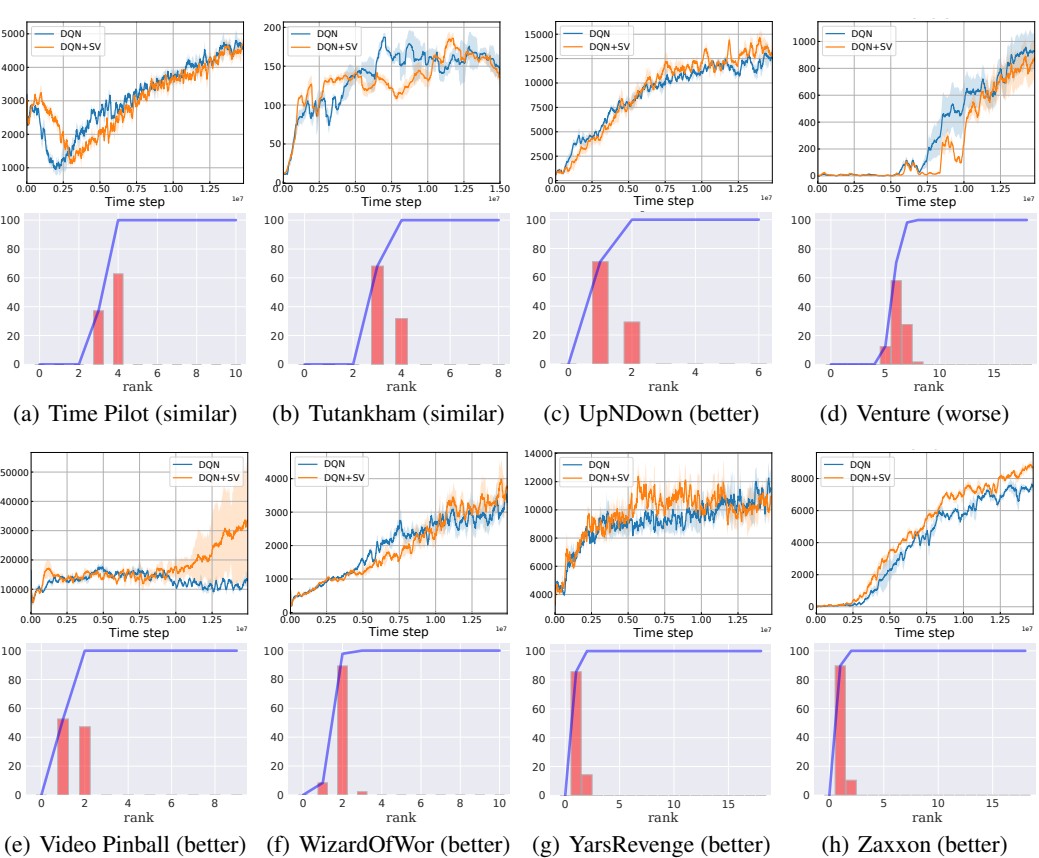

Figure 26: Additional results of SV-RL on DQN (Part D).

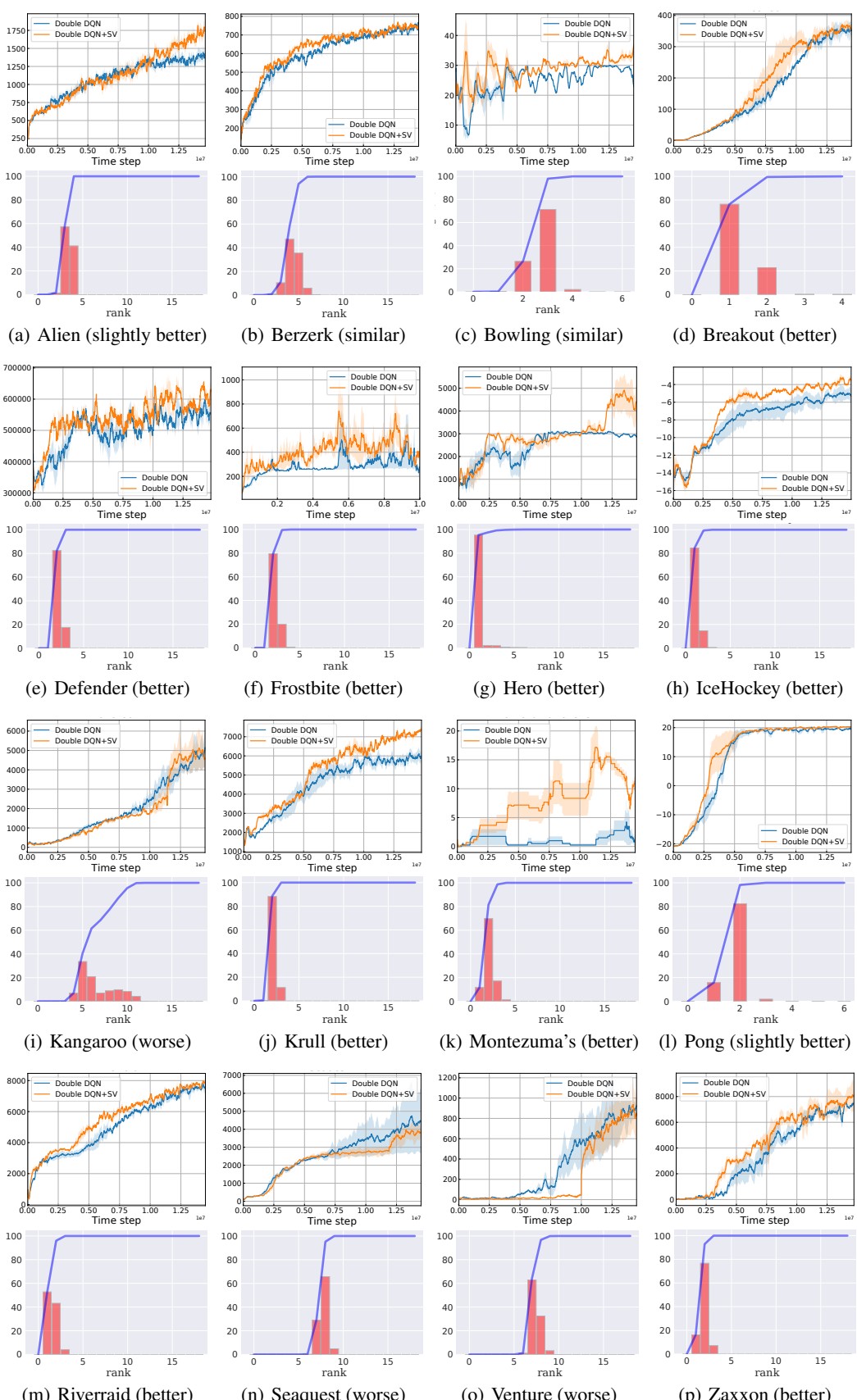

Figure 27: Additional results of SV-RL on double DQN.

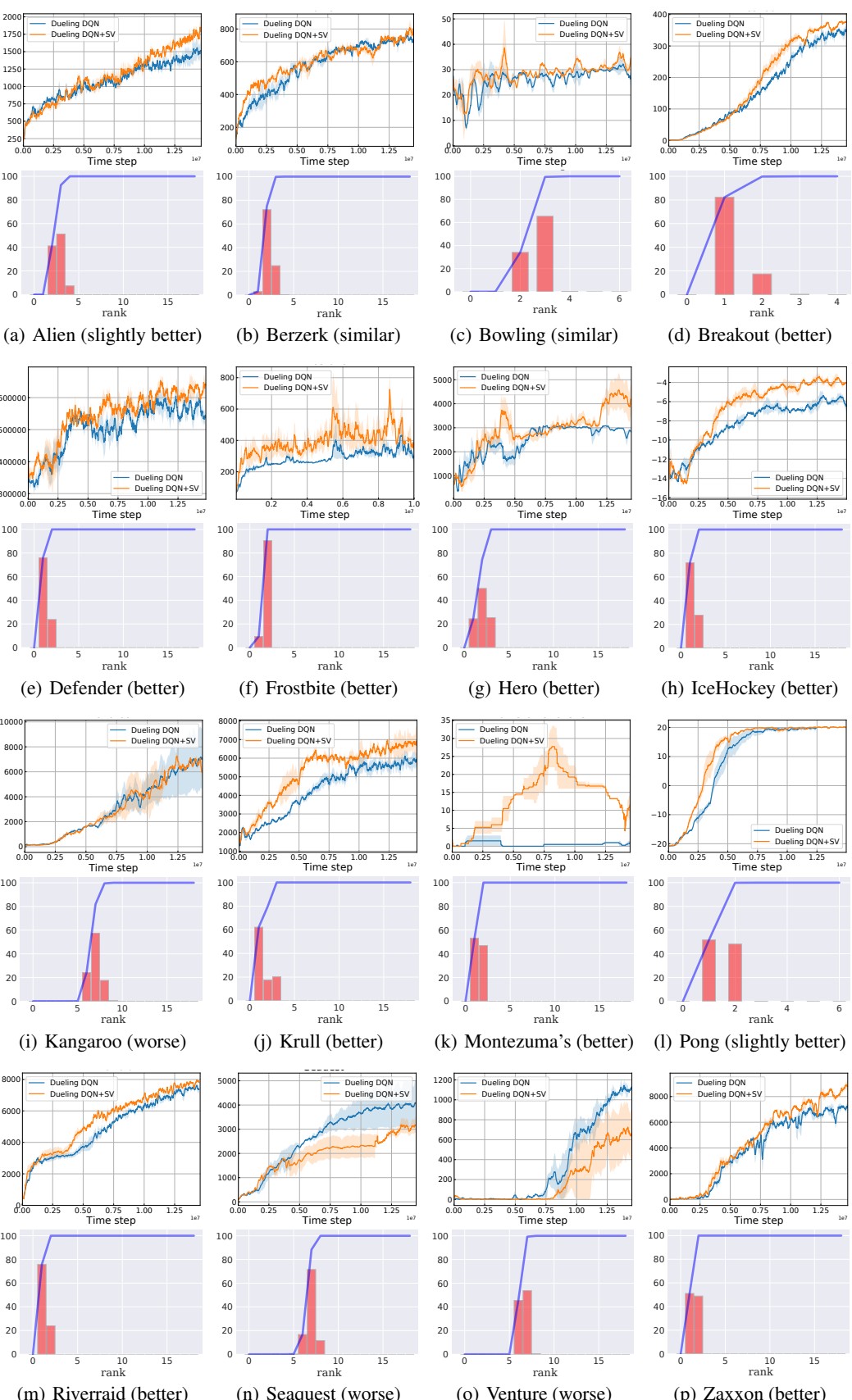

Figure 28: Additional results of SV-RL on dueling DQN.

# F  ADDITIONAL DISCUSSIONS ON RELATED WORK

**Different Structures in MDP** We note that there are previous work on exploiting low-rank structure of the system dynamics, in particular, in model learning and planning for predictive state representation (PSR). PSR is an approach for specifying a dynamical system, which represents the system by the occurrence probabilities of future observation sequences (referred as tests), given the histories. When the prediction of any test is assumed to be a linear combination of the predictions of a collection of core tests, the resulting system dynamics matrix (Singh et al., 2012), where the entries are the predictions of the core tests, is low-rank. Several extensions and variations are explored (Boots et al., 2011; Bacon et al., 2015; Ong et al., 2011), and spectral methods that often employ SVD are developed to learn the PSR system dynamics. In contrast, we remark that rather than exploiting the rank of a representation of system dynamics, we study the low-rank property of the $Q$ matrix. Instead of learning a model, we aim to exploit this low-rank property in the $Q$ matrix to develop planning and model-free RL approaches via ME.

In this work, we focus on the global, linear algebraic structure of the $Q$-value matrices, the rank, to develop effective planning and RL methods. We remark that there are studies that explore different "rank" properties of certain RL quantities, other than the $Q$-value matrices, so as to exploit such properties algorithmically. For instance, (Jiang et al., 2017) defines a new complexity measure of MDPs referred as the Bellman rank. The Bellman rank captures the interplay between the MDPs and a given class of value-function approximators. Intuitively, the Bellman rank is a uniform upper bound on the rank of a certain collection of Bellman error matrices. As a consequence, the authors develop an algorithm that expedites learning for tasks with low Bellman rank. As another example, in multi-task reinforcement learning with sparsity, (Calandriello et al., 2014) develops an algorithm referred as feature learning fitted $Q$-iteration which employs low-rank property of the weight matrices of tasks. It assumes that all the tasks can be accurately represented by linear approximation with a very small subset of the original, large set of features. The author relates this higher level of shared sparsity among tasks to the low-rank property of the weight matrices of tasks, through an orthogonal transformation. As a result, this allows the author to solve a nuclear norm regularized problem during fitted $Q$-iteration.

**MDP Planning** In terms of MDP planning, we note that there is a rich literature on improving the efficiency of planning that trades off optimality, such as the bounded real-time dynamic programming method (BRTDP) (McMahan et al., 2005) and the focused real-time dynamic programming method (FRTDP) (Smith & Simmons, 2006). Both methods attempt to speed up MDP planning by leveraging information about the value function. In particular, they keep upper and lower bounds on the value function and subsequently, use them to guide the update that would focus on states that are most relevant and poorly understood. To compare, no bounds of the value function are used to guide the value iteration in our approach. We explicit use the low-rank structure of the $Q$ matrix. Some entries (state-action pairs) of $Q$ matrix are randomly selected and updated, while the rest of them are completed through ME.

## G  ADDITIONAL EMPIRICAL STUDY

### G.1  DISCRETIZATION SCALE ON CONTROL TASKS

We provide an additional study on the Inverted Pendulum problem with respect to the discretization scale. As described in Sec. 3, the dynamics is described by the angle and the angular speed as $s = (\theta, \dot{\theta})$, and the action $a$ is the torque applied. To solve the task with value iteration, the state and action spaces need to be discretized into fine-grids. Previously, the two-dimensional state space was discretized into 50 equally spaced points for each dimension and the one-dimensional action space was evenly discretized into 1000 actions, leading to a $2500 \times 1000$ $Q$-value matrix. Here we choose three different discretization values for state-action pairs: (1) $400 \times 100$, (2) $2500 \times 1000$, and (3) $10000 \times 4000$, to provide different orders of discretization for both state and action values.

As Table 1 reports, the approximate rank is consistently low when discretization varies, demonstrating the intrinsic low-rank property of the task. Fig. 29 and Table 1 further demonstrates the effectiveness of SVP: it can achieve almost the same policy as the optimal one even with only 20% observations. The results reveal that as long as the discretization is fine enough to represent the optimal policy for the task, we would expect the final $Q$ matrix after value iteration to have similar rank.

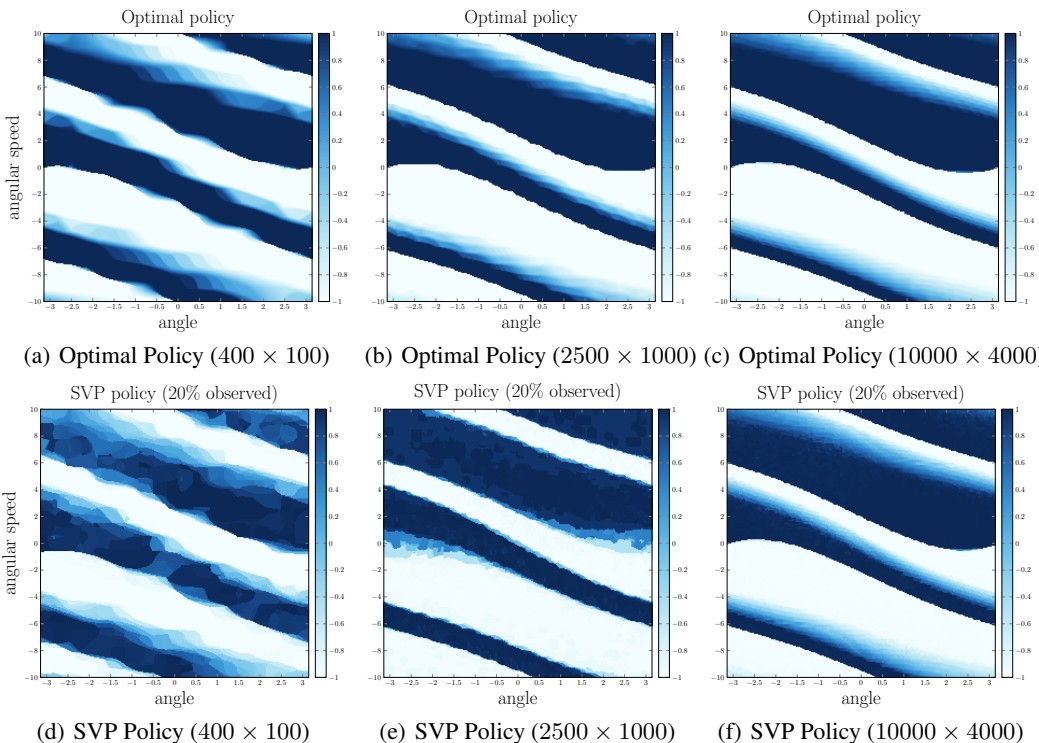

(a) Optimal Policy ($400 \times 100$)  (b) Optimal Policy ($2500 \times 1000$)  (c) Optimal Policy ($10000 \times 4000$)

(d) SVP Policy ($400 \times 100$)  (e) SVP Policy ($2500 \times 1000$)  (f) SVP Policy ($10000 \times 4000$)

Figure 29: **Additional study on discretization scale.** We choose three different discretization value on the Inverted Pendulum task, i.e. 400 (states, 20 each dimension) $\times$ 100 (actions), 2500 (states, 50 each dimension) $\times$ 1000 (actions), and 10000 (states, 100 each dimension) $\times$ 4000 (actions). First row reports the optimal policy, second row reports the SVP policy with 20% observation probability.

| Discretization Scale | Approximate Rank | Average deviation (degree) | |
|:---:|:---:|:---:|:---:|
| | | Optimal Policy | SVP Policy |
| $400 \times 100$ | 4 | 1.49 | 2.07 |
| $2500 \times 1000$ | 7 | 0.53 | 1.92 |
| $10000 \times 4000$ | 8 | 0.18 | 0.96 |

Table 1: **Additional study on discretization scale (cont.).** We report the approximate rank as well as the performance metric (i.e., the average angular deviation) on different discretization scales.

### G.2    BATCH SIZE ON DEEP RL TASKS

To further understand our approach, we provide another study on batch size for games of different rank properties. Two games from Fig. 7 are investigated; one with a small rank (Frostbite) and one with a high rank (Seaquest). Different batch sizes, 32, 64, and 128, are explored and we show the results in Fig. 30.

Intuitively, for a learning task, the more complex the learning task is, the more data it would need to fully learn the characteristics. For a complex game with higher rank, a small batch size may not be sufficient to capture the game, leading the recovered matrix via ME to impose a structure that deviates from the original, more complex structure of the game. In contrast, with more data, i.e., a larger batch size, the ME oracle attempts to find the best rank structure that would effectively describe the rich observations and at the same time, balance the reconstruction error. Such a structure is more likely to be aligned with the underlying complex task. Indeed, this is what we observe in Fig. 30. As expected, for Seaquest (high rank), the performance is worse than the vanilla DQN when the batch size is small. However, as the batch size increases, the performance gap becomes smaller, and eventually, the performance of SV-RL is the same when the batch size becomes 128. On the other hand, for games with low rank, one would expect that a small batch size would be enough to explore the underlying structure. Of course, a large batch size would not hurt since the game is intrinsically low-rank. In other words, our intuition would suggest SV-RL to perform better across different batch sizes. Again, we observe this phenomenon as expected in Fig. 30. For Frostbite (low rank), under different batch sizes, vanilla DQN with SV-RL consistently outperforms vanilla DQN by a certain margin.

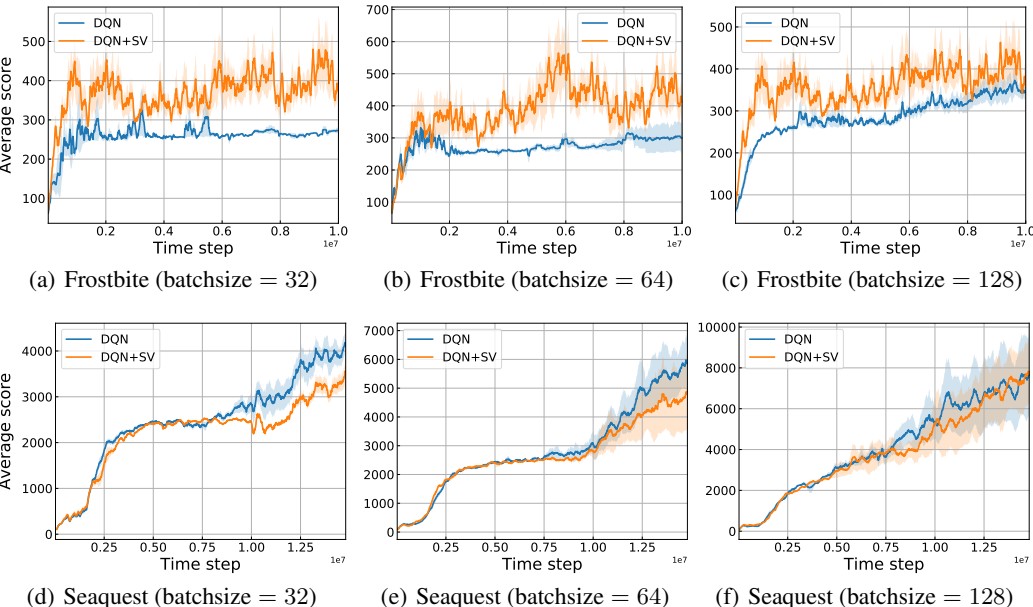

(a) Frostbite (batchsize = 32)       (b) Frostbite (batchsize = 64)       (c) Frostbite (batchsize = 128)

(d) Seaquest (batchsize = 32)       (e) Seaquest (batchsize = 64)       (f) Seaquest (batchsize = 128)

Figure 30: **Additional study on batch size.** We select two games for illustration, one with a small rank (Frostbite) and one with a high rank (Seaquest). We vary the batch size with 32, 64, and 128, and report the performance with and without SV-RL.

