# OpenReview forum: "Harnessing Structures for Value-Based Planning and Reinforcement Learning"
_ICLR.cc/2020/Conference — Accept (Talk)_

### Official Review · AnonReviewer3 · 2019-10-17
**Official Blind Review #3**

**Rating:** 8

**Review:**

Summary:
	This paper develops a method for taking advantage of structure in the value function to facilitate faster planning and learning. The key insight is that MDPs with low rank Q^* matrices can be solved more expediently using matrix estimation methods, both for classical dynamic programming methods (value iteration) and for learning in rich environments using recent model-free deep RL techniques. Thorough empirical analysis is conducted both for value iteration in tabular MDPs and for deep RL in rich environments. These experiments highlight new findings about the role the rank of the Q matrix plays in planning convergence and learning rates.

	I view this paper as containing several key contributions: first, the analysis on the role of Q rank in planning and learning---experiments conducted indicate that even complicated environments tend to have low rank Q matrices (when approximated). Highlighting the role of this rank and the corresponding empirical analysis estimating it in benchmark RL and control tasks is, to my knowledge, novel. Second, and perhaps the most significant contribution, is "Structured Value RL" (SV-RL), an easy-to-apply method that can be incorporated into many Q-based deep RL methods with little overhead. The empirical results are compelling: across three different variations of DQN-like architectures, the SV RL augmentation tends to improve learning. Presentation of results is rigorous, too, and provide strong evidence that the method works.

	As the paper mentions, theoretical analysis on the impact of Q-rank on dynamic programming (and perhaps learning) would be of great interest to the community. I take this analysis to be out of scope for this paper, but could see the work motivating future investigation into these questions.

Verdict: Overall, I take this paper to present many novel insights, establish solid motivation with good writing and examples, and offers compelling evidence about the strength of SV RL. I recommend accepting the paper.

Comments:
	C1: The visuals throughout the paper are helpful!
	C2: The paper is well written: the use of examples was effective in developing the motivation.
	C3: Section 2 is helpful for understanding the ideas developed in the paper. However, there are many well developed planning frameworks for MDPs that trade-off optimality with computational efficiency. It might be worth discussing some of these methods up front. For instance, Bounded Real-Time Dynamic Programming (McMahan et al. 2005) explicitly uses value function structure to improve planning speed, with performance guarantees, as does Focused RTDP (Smith and Simmons 2006). I don't take the computational complexity improvements of the proposed method to be the primary contribution, so just a brief discussion to contextualize the work against other planning literature would be helpful.
	C4: While the "rank" studied here is of a different form, some discussion of the Bellman Rank work (Jiang et al. 2017) might be useful for differentiating the two notions of "rank" at play, and how they are each used to expedite learning. The Bellman Rank is used as a measure of complexity of an MDP---Jiang et al. develop an RL algorithm that has sample complexity that depends on this measure. It is not strictly necessary, but I could see multiple uses of "rank" appearing in the RL literature as a means of exploiting structure for faster learning being confusing. If space (perhaps in the appendix if not), a sentence or two differentiating the two ideas might be helpful to readers. Additionally, the study of sparsity in value function representation was studied by Calandriello et al. 2014. If space permits, the paper might benefit from some discussion of the relation to this work.

Questions:
	Q1: In the inverted pendulum results, I am curious about the effect of the discretization on plan quality. Specifically: how were 2500 states and 1000 actions chosen? Were different orders of magnitude (for both values) considered? How did this impact SVP? Does the rank change as the discretization becomes more or less coarse? I don't think this is strictly critical for the paper, but a few sentences clarifying this point would be informative.

	Q2: Figure 4 provides nice insights into how to scale these ideas to deep RL. How were the four games chosen? Is there anything special that motivated their selection?

	Q3: Additionally, I am curious about whether the results from Figure 4 are the consequence of algorithmic decisions, rather than the environment. Is it possible to determine whether different value based methods (or different choices of hyperparameters) lead to different outcomes? For instance, I could imagine a more shallow network, or a tighter bottleneck, leading to Q evaluations that produce higher rank.


Typos and Writing Suggestions:

[Abstract]
	- This sentence is quite long, and I had a hard time following it as a result: "As our key contribution, by leveraging...". Consider dividing into two sentences.

[Intro]
	- Oxford comma: "control, planning and reinforcement learning"::"control, planning, and reinforcement learning
	- "the structured dynamic"::"the structure in the dynamics"
	- "where much fewer samples"::"where fewer samples"
	- Consider rewording: "almost the same policy as the optimal one". Is it that the policies are in fact the same? Or that their values are close? Perhaps: "a policy with near optimal value".
	- When introducing Double DQN and Dueling DQN for the first time it would be appropriate to cite each (end of Section 1).

[Sec. 2: Warm Up]
	- "understand the structures"::"understand the structure"
	- "give a strong evidence for"::"provide evidence that"
	- "exploit the structures for"::"exploit structure in the value function for"
	- I think the italicized statement at the top of page 3 could be sharpened. The antecedent currently stating "why not" is quite a soft statement compared to the motivation the section develops. Consider changing: "...why not enforcing such a structure throughout the iterations?"::"...then enforcing such a structure throughout planning can improve the rate of convergence".

[Sec. 3: Structured ... Planning]
	- "even non-convex optimization approaches (..."::"even non-convex optimization approaches to solving this problem (..."
	- "offer a sounding foundation for future"::"offer a sound foundation for future"

[Sec. 4: Structured ... RL]
	- "Previously, we start by"::"Previously, we started by"
	- "which in deep scenarios"::"which in scenarios with large state spaces", or perhaps: "which in deep scenarios"::"which in scenarios where value function approximation is used"


References:

Calandriello, Daniele, Alessandro Lazaric, and Marcello Restelli. "Sparse multi-task reinforcement learning." Advances in Neural Information Processing Systems. 2014.

Jiang, Nan, et al. "Contextual decision processes with low Bellman rank are PAC-learnable." Proceedings of the 34th International Conference on Machine Learning-Volume 70. JMLR. org, 2017.

McMahan, H. Brendan, Maxim Likhachev, and Geoffrey J. Gordon. "Bounded real-time dynamic programming: RTDP with monotone upper bounds and performance guarantees." Proceedings of the 22nd international conference on Machine learning. ACM, 2005.


Smith, Trey, and Reid Simmons. "Focused real-time dynamic programming for MDPs: Squeezing more out of a heuristic." AAAI. 2006.


**Experience Assessment:**

I have published one or two papers in this area.

**Review Assessment: Checking Correctness Of Derivations And Theory:**

I assessed the sensibility of the derivations and theory.

**Review Assessment: Checking Correctness Of Experiments:**

I assessed the sensibility of the experiments.

**Review Assessment: Thoroughness In Paper Reading:**

I read the paper thoroughly.

---

> ### Author Response · Authors · 2019-11-12
> **Response to Review #3 (part 1/2)**
>
> Thank you very much for acknowledging the novelty and the contributions of our work! We are delighted to see that you enjoyed reading the paper, and that the examples and experiments were effective in developing the intuition. In addition, we thank you for your efforts in evaluating the paper and provide many useful writing suggestions. These are important for us to further improve the presentation of this paper. In the following, we provide details for your comments and questions.
>
> [Comments]
> C3 & C4: Thank you for the reference. These are useful for us to provide a more informed context of our work. Due to space limit, we provide below the text that we plan to add in the revised manuscript.
>
> In terms of MDP planning, we note that there is a rich literature on improving the efficiency of planning that trades off optimality, such as the bounded real-time dynamic programming method (BRTDP) [1] and the focused real-time dynamic programming method (FRTDP) [2].  Both methods attempt to speed up MDP planning by leveraging information about the value function. In particular, they keep upper and lower bounds on the value function and subsequently, use them to guide the update that would focus on states that are most relevant and poorly understood. To compare, no bounds of the value function are used to guide the value iteration in our approach. We explicit use the low-rank structure of the Q matrix. Some entries (state-action pairs) of Q matrix are randomly selected and updated, while the rest of them are completed through ME.
>
> [1] McMahan, H. Brendan, Maxim Likhachev, and Geoffrey J. Gordon. "Bounded real-time dynamic programming: RTDP with monotone upper bounds and performance guarantees." Proceedings of the 22nd international conference on Machine learning. ACM, 2005.
> [2] Smith, Trey, and Reid Simmons. "Focused real-time dynamic programming for MDPs: Squeezing more out of a heuristic." AAAI. 2006.
>
> In this work, we focus on the global, linear algebraic structure of the Q-value matrices, the rank, to develop effective planning and RL methods. We remark that there are studies that explore different ``rank” properties of certain RL quantities, other than the Q-value matrices, so as to exploit such properties algorithmically. For instance, [1] defines a new complexity measure of MDPs referred as the Bellman rank. The Bellman rank captures the interplay between the MDPs and a given class of value-function approximators. Intuitively, the Bellman rank is a uniform upper bound on the rank of a certain collection of Bellman error matrices. As a consequence, the authors develop an algorithm that expedites learning for tasks with low Bellman rank. As another example, in multi-task reinforcement learning with sparsity, [2] develops an algorithm referred as feature learning fitted Q-iteration which employs low-rank approximation techniques. It assumes that all the tasks can be accurately represented by linear approximation with a very small subset of the original, large set of features. The author relates this higher level of shared sparsity among tasks to the low-rank property of the weight matrices of tasks, through an orthogonal transformation. As a result, this allows the author to solve a nuclear norm regularized problem during fitted Q-iteration.
>
> [1] Jiang, Nan, et al. "Contextual decision processes with low Bellman rank are PAC-learnable." Proceedings of the 34th International Conference on Machine Learning-Volume 70. JMLR. org, 2017.
> [2] Calandriello, Daniele, Alessandro Lazaric, and Marcello Restelli. "Sparse multi-task reinforcement learning." Advances in Neural Information Processing Systems. 2014.

---

> > ### Author Response · Authors · 2019-11-12
> > **Response to Review #3 (part 2/2)**
> >
> > [Questions]
> > Q1). In general, for continuous control, the quality of policy can be affected by the discretization. The discretization should be fine enough so as to capture the system dynamics. In our case, the two-dimensional state space was discretized into 50 equally spaced points for each dimension and the one-dimensional action space was evenly discretized into 1000 actions. We observed that it gave smooth enough optimal policy as shown in the paper, and hence, was fixed throughout. Discretization affects the size of the Q matrix, which might slightly affect the approximate rank. However, an intrinsically low-rank task should stay low-rank: as long as the discretization is fine enough to represent the optimal policy for the task, we would expect the final Q matrix after value iteration to have similar rank, and SVP to effectively learn an approximately optimal policy with less updates.
> >
> > We have added a new additional study in Appendix F.1 to address the effect of discretization. We would like to refer you to the newly added section, where we provide additional experiments and visualizations of different discretization scales to support our statements.
> >
> > Q2). In fact, the rank plots of DQN for all the 57 games are provided in Appendix E (Figures 23 - 26). We observe that many games do exhibit a low-rank structure.To illustrate the intuition, four representative games are selected and results are presented in Figure 4.
> >
> > Q3). This is a good point. We believe that these are the consequences of environments, rather than algorithms. In fact, we tried to study the effect of different value-based methods. The results are provided in Figures 27 and 28 in Appendix E. There, we show the rank plots of Double DQN and Dueling DQN, for 16 games. We observe that the results are aligned with those presented in Figures 23 - 26 for DQN. The rank might change slightly for different methods, but the overall behavior are more or less the same — low-rank tasks stay low-rank across different methods, while the rank of complex tasks are consistently high.
> >
> > As for the network, we use standard architectures as in the previous work (i.e., the original DQN). These networks provide very high degree of freedom and produce great results in the literature. As such, we do not believe that they are the limiting factor for our rank study. As illustrated in Figures 23 - 26, many games have small ranks such as 4 or 5. Moreover, the same network leads to both high-rank and low-rank results, depending on the games. That is, given an expressive enough network, the rank should not be restricted by the neural network, but rather depending on the environments.
> >
> > To summarize, we believe that the above results give evidence that the rank behavior is determined by the characteristics of the environments.
> >
> >
> > [Typos and Writing Suggestions]
> > Thank you for your constructive feedback. We really appreciate them! We will definitely revise the manuscript to incorporate your detailed suggestions.

---

> > > ### Comment · AnonReviewer3 · 2019-11-13
> > > **R3 Response to Rebuttal**
> > >
> > > I thank the authors for their thorough response, clarifications, and additional experiments.
> > >
> > > Re Q1): The results presented in Appendix F1 provide a satisfying investigation into the effect of the discretization.
> > >
> > > Re Q2): Including results for dueling/double DQN is very helpful. I take the additional results (Fig. 24-28) as supporting the main claim that, on low rank games, +SV consistently helps, whereas in higher rank games, there is higher variance to the effect of +SV.
> > >
> > > Re Q3:) Again, these additional results answer my question; there is little variation in rank estimation across learning algorithms.
> > >
> > > I take the paper to be strengthened by the additional results and discussion, and still recommend acceptance.

---

### Official Review · AnonReviewer1 · 2019-10-22
**Official Blind Review #1**

**Rating:** 8

**Review:**

The study is motivated by the observation that the Q-value matrix in reinforcement learning problems often has a low-rank structure. The paper proposes an approach called structured value-based planning or learning, where the Q matrix or the Q function is estimated from incomplete observations based on the prior that it is low-rank. The proposed strategy is demonstrated in stochastic control tasks and reinforcement learning applications.

The paper is clearly written and the experimental results show that the proposed strategy leads to performance gains especially in problems where the Q matrix indeed conforms to a low-rank model. A few comments and questions:

- The assumption that the Q matrix should be low-rank is demonstrated with several experiments. Is there any theoretical motivation or guarantee for this assumption as well?

- The experimental results show that the proposed strategy performs well in problems that are low-rank, while the performance may degrade in problems where the low-rank assumption is not met. Would it be possible to detect the rank of the problem in a dynamical manner (i.e., during the learning), so that the number of incomplete observations of Q can be increased to improve the performance, or the solution strategy (e.g. whether to use the low-rank assumption or not) can be adapted to the nature of the problem?

- The Q-value matrices and functions considered in the problem have a special structure as they result from Markov Decision Processes. Would it be possible to go beyond the low-rank assumption and propose and use a more elaborate type of prior that employs the special structure of MDPs?

- Please clearly define the notation used in Section 4.2.

**Experience Assessment:**

I do not know much about this area.

**Review Assessment: Checking Correctness Of Derivations And Theory:**

I assessed the sensibility of the derivations and theory.

**Review Assessment: Checking Correctness Of Experiments:**

I assessed the sensibility of the experiments.

**Review Assessment: Thoroughness In Paper Reading:**

I made a quick assessment of this paper.

---

> ### Author Response · Authors · 2019-11-12
> **Response to Review #1**
>
> Thank you very much for your supportive remark! We are happy that the writing is clear to you. Below we provide additional comments regarding your questions.
>
> 1). Low-rank assumption:
> This work was primarily motivated by the observation that many systems exhibit strong relationship among states and actions, governed by potentially simple dynamics. This might eventually lead to structures within the optimal solution. We hope that this empirical study would motivate further theoretical analysis on structures within the community. Below are some thoughts on the potential theoretical motivation for this assumption:
>
> a) It is possible that the states and actions in consideration have some latent variable representations. If the optimal Q function is a piecewise analytic function on the latent variables, then there are works arguing the approximately low-rank property of the resulting matrix [1].
> b) There are theoretical works in RL and Markov process that assume that the transition kernel can be decomposed to a low-dimensional feature representation [2,3]. These assumptions on the transition kernel may lead to low-rank optimal Q matrices.
> c) For continuous problems, theoretical analysis often needs to assume some sort of smoothness in the Q function [4,5]. It is possible that such smoothness in the Q function will result in a low-rank Q matrix when evaluated at finite but fine enough discretized grid.
>
> [1] Udell, Madeleine, and Alex Townsend. "Why Are Big Data Matrices Approximately Low Rank?." SIAM Journal on Mathematics of Data Science 1.1 (2019): 144-160.
> [2] Yang, Lin, and Mengdi Wang. "Sample-Optimal Parametric Q-Learning Using Linearly Additive Features." International Conference on Machine Learning. 2019.
> [3] Sun, Yifan, et al. "Learning low-dimensional state embeddings and metastable clusters from time series data." Neural Information Processing Systems 2019.
> [4] Yang, Zhuora, Yuchen Xie, and Zhaoran Wang. "A theoretical analysis of deep Q-learning." arXiv preprint arXiv:1901.00137(2019).
> [5] Shah, Devavrat, and Qiaomin Xie. "Q-learning with nearest neighbors." Advances in Neural Information Processing Systems. 2018.
>
>
> 2). Dynamical manner for the number of incomplete observations and whether the strategy can be adapted to the nature of the problem:
> This is a great point and definitely an interesting future direction. In the current work, it is not immediately that one could easily detect the rank and adapt the algorithm in a principled manner. As one practical solution, it may be possible to dynamically adjust the regularization in a manner similar to cross validation. At each step, for the submatrix, one could randomly sample a portion of the entries for ME, while keeping another fraction of the remaining entries as a validation set. If the recovered matrix via ME has a low reconstruction error on the validation set, it is likely that a suitable low-rank approximation is sufficient and has been found by the ME oracle. In contrast, if the reconstruction error is large, the algorithm might have been too aggressive on finding a low-rank solution while a higher rank solution is indeed necessary. As such, one could then adjust the algorithm to increase the number of observations for ME or try to reduce the level of low-rank regularization.
>
> The above cross validation scheme might be an interesting complement to our current approach. Overall, we believe that principally solving those questions you posted are meaningful and important directions that worth further investigations.
>
>
> 3). Beyond the low-rank assumption and use a more elaborate type of prior:
> Thank you for your inspiring advice. Without any elaborate prior information, rank is a natural point to study the global property of a matrix. In principle, understanding structures in MDP could also be potentially explored, and we believe that it is possible to extend to other types of scenarios with prior information about the MDPs. However, at the current stage, we do not have a particularly systematic approach to explore more elaborate type of structures in MDPs.
> While this paper is focusing on low-rank structures, as the reviewer noted, there can be other structures to be explored. We hope that our paper could serve as an example, and further motivates future studies for exploiting structures in MDP.
>
>
> 4). Notation in Section 4.2:
> Thank you for the suggestion. We will expand the definition of the notation to make them clearer.

---

> > ### Comment · AnonReviewer1 · 2019-11-14
> > **Thanks for the response**
> >
> > I would like to thank the authors for the time they took to answer my questions. They have addressed all my comments.

---

### Official Review · AnonReviewer2 · 2019-10-22
**Official Blind Review #2**

**Rating:** 6

**Review:**

This paper introduces an interesting idea of exploiting the low-rank structure of the value function to reduce the computation complexity of value-based RL algorithms. Instead of working in the reduced space, they focus to operate on the original space and reduce the computation by looking at few elements and inferring the rest.  They use a matrix completion/estimation strategy to infer the global structure from a smaller set of samples. They show empirical evidence of the low-rank structure in few classical control tasks (Mountain Car, Inverted Pendulum, Cart Pole), and provide an iterative procedure - Structure Value-based Planning (SVP), that is similar to value iteration but is able to exploit the low-rank structure to reduce the computational time.  They also provide a deep RL extension - SV-RL, that can be applied to value-based methods. They test the efficiency of their approach to Atari games.


Overall this paper presents an interesting idea, that also scales to the deep RL algorithms. However, there are a few missing components that need to be addressed in order to fully support the claims.  Given these clarifications in the author's response, I would be willing to increase the score.

1) Nature of the regularization for SV-RL.
The authors are proposing a form of regularization that enforces low-rank structure for the value function (and target Q-values in particular for deep RL agents).  The authors show that this form of regularization is helpful for improving the learning for low-rank tasks, and for tasks that have a high-rank the performance is worse.  This kind of regularization balances having a low-rank and small reconstruction error. However, shouldn’t the regularization also depend on the size of the sub-matrix (the minibatch)?  However, I didn’t find any experiments related to how changing it affects performance. Also, is the regularization related to due to the random projections (Johnson–Lindenstrauss lemma)?

2) It is important to note that SV-RL is limited to Deep Q-learning based techniques. So it can’t be applied to any value-based method, especially when the samples in the sub-matrix are correlated.

3) Missing literature that exploits Low-rank structure for planning. There is literature on RL that is based on exploiting the low-rank structure for planning [1, 2, 3].

4) All the experiments are in deterministic environments? Is there a reason behind this?

5) (Optional) This regularization introduces error in reconstructed approximate Q-values. It will be useful to have some analysis on how far it deviates from the optimal value function. There has been work in the field [4, 5] that I believe can be used to help derive an analysis of the kind of approximation error bounds that are being introduced here.



References:


[1] Byron Boots, Sajid M Siddiqi, and Geoffrey J Gordon. Closing the learning-planning loop with predictive state representations. The International Journal of Robotics Research, 30(7):954–966, 2011

[2] Pierre-Luc Bacon, Borja Balle, and Doina Precup. Learning and planning with timing information in markov decision processes. In Proceedings of the Thirty-First Conference on Uncertainty in Artificial Intelligence , pp. 111–120. AUAI Press, 2015.

[3] Sylvie CW Ong, Yuri Grinberg, and Joelle Pineau. Goal-directed online learning of predictive models. In European Workshop on Reinforcement Learning, pp. 18–29. Springer, 2011.

[4] Klopp, Olga. "Noisy low-rank matrix completion with general sampling distribution." Bernoulli 20.1 (2014): 282-303.

[5] Recht, Benjamin. "A simpler approach to matrix completion." Journal of Machine Learning Research 12.Dec (2011): 3413-3430.





**Experience Assessment:**

I have published one or two papers in this area.

**Review Assessment: Checking Correctness Of Derivations And Theory:**

N/A

**Review Assessment: Checking Correctness Of Experiments:**

I assessed the sensibility of the experiments.

**Review Assessment: Thoroughness In Paper Reading:**

I read the paper thoroughly.

---

> ### Author Response · Authors · 2019-11-12
> **Response to Review #2 (part 1/2)**
>
> Thank you very much for your positive sentiment! We are glad that you found the idea interesting. In the following, we address your concerns in detail. We hope that these will further clarify our work and lead to a favorable increase of the score.
>
> 1).
> a) Regularization and batch size: Thank you for your great point! We add additional experiments and provide an additional study on batch size in Appendix F.2. We would like to refer you to the newly added section, where different batch sizes are explored for games of different rank properties. To summarize, intuitively, the more complex the learning task is, the more data it would need to fully learn the characteristics. For a complex game with higher rank, a small batch size may not be enough to capture the game, resulting in ME to impose an inadequate structure during learning. With a larger batch size (i.e., more data), we would expect ME to find the best rank structure that is aligned with the sufficiently rich data. Therefore, the performance gap between SV-RL and vanilla DQN should decrease as the batch size increases. On the other hand, a small batch size would like to be enough for an intrinsically low-rank games, and we would expect the performance to be not dramatically affected by increasing the batch size. That is, SV-RL should consistently outperform the vanila DQN. Indeed, we observe the above desired results in our experiments. Again, a more elaborate discussion can be found in Appendix F.2.
>
> b) Johnson-Lindenstrauss Lemma: In spirit, Johnson-Lindenstrauss lemma concerns low-dimensional embedding from high-dimensional data. In fact, it could be used to argue why big data matrices, in particular, with latent variables models, are low-rank [1]. In this sense, it is related to our study of low-rank Q matrices. However, our work focuses on how to use such a structure. We would attribute the regularization to the ME’s formulation which intrinsically seeks low-rank solution that respects the observed data. Classical Johnson-Lindenstrauss lemma concerns about projecting the data to a random subspace, i.e., orthogonal projection onto a random subspace in R^n uniformly distributed in the Grassmannian (Section 5.3 of [2]). In contrast, ME is more direct and explicit. Rather than a random subspace, we can view ME as explicitly projecting onto a space with “low” rank that is mostly aligned with the observed entries (data). As such, we believe that it naturally regularize the solution to have the low-rank property.
>
> [1] Udell, Madeleine, and Alex Townsend. "Why Are Big Data Matrices Approximately Low Rank?." SIAM Journal on Mathematics of Data Science 1.1 (2019): 144-160.
> [2] Vershynin, Roman. “High-dimensional probability: An introduction with applications in data science.” Vol. 47. Cambridge University Press, 2018.
>
>
> 2). Thank you for your point! In our current presentation for deep RL (Section 4), we agree that SV-RL is limited to deep Q-learning based techniques. In fact, this was made clear in Section 4.2 (last paragraph of page 5): “It exploits the structure for the sampled batch at each SGD step, and can be easily incorporated into any Q-value based RL methods that update the Q network via a similar step as in Q-learning.” We will rephrase the wording in the introduction to make this point clear.
>
> Relatedly, we believe that this does not limit the value of our work. Deep Q-learning is one of the most fundamental and important value based approaches, which has been widely used and extended in a variety of tasks. Therefore, we believe our approach has a broad interest in many fields, and additionally, could motivate the study of structure for other RL methods.

---

> > ### Author Response · Authors · 2019-11-12
> > **Response to Review #2 (part 2/2)**
> >
> > 3). Thank you for the reference. We will update the related work in the revised version as follows.
> >
> > We note that there are previous works on exploiting low-rank structure of the system dynamics, in particular, in model learning and planning for predictive state representation (PSR). PSR is an approach for specifying a dynamical system, which represents the system by the occurrence probabilities of future observation sequences (referred as tests), given the histories. When the prediction of any test is assumed to be a linear combination of the predictions of a collection of core tests, the resulting system dynamics matrix [1] , where the entries are the predictions of  the core tests, is low-rank. Several extensions and variations are explored [2,3,4], and spectral methods that often employ SVD are developed to learn the PSR system dynamics. In contrast, we remark that rather than exploiting the rank of a representation of system dynamics, we study the low-rank property of the Q matrix. Instead of learning a model, we aim to exploit this low-rank property in the Q matrix to develop planning and model-free RL approaches via ME.
> >
> > [1] Singh, Satinder, Michael R. James, and Matthew R. Rudary. "Predictive state representations: A new theory for modeling dynamical systems." Proceedings of the 20th conference on Uncertainty in artificial intelligence. AUAI Press, 2004.
> > [2] Byron Boots, Sajid M Siddiqi, and Geoffrey J Gordon. Closing the learning-planning loop with predictive state representations. The International Journal of Robotics Research, 30(7):954–966, 2011
> > [3] Pierre-Luc Bacon, Borja Balle, and Doina Precup. Learning and planning with timing information in markov decision processes. In Proceedings of the Thirty-First Conference on Uncertainty in Artificial Intelligence , pp. 111–120. AUAI Press, 2015.
> > [4] Sylvie CW Ong, Yuri Grinberg, and Joelle Pineau. Goal-directed online learning of predictive models. In European Workshop on Reinforcement Learning, pp. 18–29. Springer, 2011.
> >
> >
> > 4). To clarify, two of the selected planning tasks are actually non-deterministic. For the Inverted Pendulum and the Cart-Pole, we add a Gaussian noise when updating the angular speed term for the pendulum and the pole in Eq. (6) & (17). Such operations are standard and common in many benchmarks such as OpenAI-Gym, which we follow to implement our dynamics. In addition, the initial state of all the control tasks are randomly selected during evaluation.
> >
> > As for Atari games, this is the standard benchmark for reporting deep RL results which has been used extensively in the literature. It happens in coincidence that those games are deterministic.
> >
> > To summarize, those environments were selected because they are classical examples and standard benchmarks for control and deep RL. This is aligned with most past literature and allows us to compare our results with them.
> >
> >
> > 5). Thank you for the reference. A theoretical analysis would definitely be helpful for understanding our approach. Indeed, we tried to address this and provided a short discussion in Appendix A on the technical difficulty for theoretical analysis. Due to space limitations, this section was unfortunately moved to the Appendix in the submitted manuscript. To summarize our answer, basically, standard ME analysis (including the ones you mentioned) focuses on the one-shot setting of recovering a static data matrix given incomplete observations, under the Frobenius norm guarantees. We agree that if under this setting (i.e., recover Q^* through noisy observation of entries of Q^* directly), there are theoretical results that we could adapt to give certain bounds. However, the iterative nature of the algorithm and need for non-trivial infinite norm bound leads to unique difficulties, which to our best knowledge, requires potentially new machinery. We hope and believe that this work would motivate future investigation into these questions.

---

### Author Response · Authors · 2019-11-12
**General Response: additional studies and a new section**

We thank all the reviewers for their insightful comments, suggestions and questions!

To better address some of the comments, we performed additional experimental studies and added a new section, Appendix F, in the updated manuscript. In Appendix F.1, additional study on discretization scale for control tasks is provided (Review #3). For Appendix F.2, we investigated the effect of batch size for deep RL tasks that have different rank properties (Review #2).

Regarding suggestions on the related work and reference, we responded to each reviewer individually, providing the text we plan to add to the final manuscript. Due to page limit, these were not updated in the current version, but will definitely be incorporated at appropriate sections in the final manuscript.

Finally, point-to-point responses can be found after each review. We hope that these address your concerns and would be happy to answer any further questions.

---

### Decision · Program_Chairs · 2019-12-19

**Decision:**

Accept (Talk)

**Comment:**

The paper shows empirical evidence that the the optimal action-value function Q* often has a low-rank structure. It uses ideas from the matrix estimation/completion literature to provide a modification of value iteration that benefits from such a low-rank structure.
The reviewers are all positive about this paper. They find the idea novel and the writing clear.
There have been some questions about the relation of this concept of rank to other definitions and usage of rank in the RL literature.
The authors’ rebuttal seem to be satisfactory to the reviewers. Given these, I recommend acceptance of this paper.